# Fantastic Gains and Where to Find Them: On the Existence and Prospect of General Knowledge Transfer between Any Pretrained Model

**Karsten Roth**[1,*], **Lukas Thede**[1,*], **A. Sophia Koepke**[1]
**Oriol Vinyals**[2], **Olivier Hénaff**[2], **Zeynep Akata**[1,3,4]
[1]Tübingen AI Center & University of Tübingen, [2]Google DeepMind
[3]Helmholtz Munich, [4]Technical University of Munich
[*]equal contribution

## Abstract

Training deep networks requires various design decisions regarding for instance their architecture, data augmentation, or optimization. In this work, we find these training variations to result in networks learning **unique** feature sets from the data. Using public model libraries comprising thousands of models trained on canonical datasets like ImageNet, we observe that for arbitrary pairings of pretrained models, one model extracts significant data context unavailable in the other – independent of overall performance. Given **any arbitrary pairing of pretrained models** and no external rankings (such as separate test sets, e.g. due to data privacy), we investigate if it is possible to transfer such "complementary" knowledge from one model to another without performance degradation – a task made particularly difficult as additional knowledge can be contained in stronger, equiperformant or weaker models. Yet facilitating robust transfer in scenarios agnostic to pretrained model pairings would unlock **training guidance, auxiliary gains and knowledge fusion** from any model repository without restrictions on model & problem specifics - including from **weaker, lower-performance models**. This work provides a first, in-depth exploration on the viability of such **general-purpose knowledge transfer**. Across large-scale experiments, we first reveal the shortcomings of standard knowledge distillation techniques, and then propose a general extension via data partitioning for successful transfer between nearly all pretrained models - which can also be done **unsupervised**. Finally, we assess both the scalability and impact of model properties on successful model-agnostic knowledge transfer.

## 1 Introduction

Training neural networks on specific datasets has become a machine learning standard to tackle a myriad of research and industry challenges, involving a large number of explicit and implicit decisions that range from architecture choices to specific optimization protocols, the particular choice of data augmentation, data sampling and even the data ordering. All these factors can impact the semantic knowledge a model can extract from a dataset (Bouthillier et al., 2021; Schmidt et al., 2021; Wang et al., 2023; Raghu et al., 2021; Wagner et al., 2022; Roth et al., 2020; Balestriero et al., 2023; Teney et al., 2020; Roth et al., 2023), and together provide a unique fingerprint of a model's capabilities. In this work, we first highlight the extent of this statement through extensive experiments. We build on large open model libraries (e.g. `timm` (Wightman, 2019) or `huggingface`) to compare large numbers of *arbitrary pretrained model pairs*. Doing so, we discover the consistent existence of significant complementary knowledge - information about the data that one model (the "teacher") holds that is not available in the other one (the "student"). Interestingly, we find that **complementary knowledge** exists *regardless* of external performance rankings or factors like model families (CNNs (LeCun and Bengio, 1995), Transformer (Dosovitskiy et al., 2021), MLP (Tolstikhin et al., 2021)), and often aggregates in semantic areas of expertise: For stronger, but especially also similar or weaker teachers (by some test metric), significant knowledge about the data *not available* to the student can be found.

Such **general availability of complementary knowledge** raises questions about its potential utility. To answer those, we provide a first, in-depth exploration. Specifically, *given arbitrary pairs of models*

*pretrained on the same data without access to external ranking measures (s.a. test sets, due to e.g. data privacy, separate test servers, ...), we explore if transfer of complementary knowledge between **any** teacher and student is possible **without** performance degradation.* Achieving such transfer through any possible model pair unlocks any freely available or self-generated model collection as an auxiliary resource for gains in canonical and problem-specific pretraining. It also avoids the need for model-specific transfer that require expert knowledge, and reduces the reliance on external evaluation measures for model selection. More importantly however, it also enables improvements of larger models by knowledge transfer from weaker, lower-resource models, without the explicit need for *additional data & supervision*, or sacrifices in e.g. speed, fairness, interpretability or ease-of-use.

We investigate the limits of knowledge distillation (Hinton et al., 2015; Tian et al., 2020) for this task, which in contrast to data-free approaches (e.g. Wortsman et al. (2022a)), operates independently of model choices. However, standard knowledge distillation frameworks assume information to be distilled to an *untrained* student. In contrast, we only wish to transfer knowledge not available in an already trained student model, which may even outperform its teacher. This crucially entails a successful trade-off between knowledge gain and retention. Indeed, for knowledge transfer between arbitrary pretrained models, common distillation (§5.1) exhibits strong model/hyperparameter dependence and performance drops for the majority of student models, particularly for weaker/equiperformant teachers. This can be attributed to catastrophic forgetting (Kirkpatrick et al., 2016; Zenke et al., 2017) outweighing the benefits of complementary knowledge transfer from the teacher.

For a favorable trade-off between forgetting and knowledge gain, we treat the transfer process as a continual learning problem, where a model is continuously presented with new context for data already seen. To encourage retention, we first study weight interpolation (Stojanovski et al., 2022; Wortsman et al., 2022b). While better than normal distillation, it is often *too strong* a constraint when the teachers have niche areas of expertise or are overall stronger. We thus propose to constrain distillation at the data level by partitioning it into two sets - one with samples where transfer from a teacher is desired, and one where we wish to retain the student behavior. This introduces significantly fewer constraints on the model weights to learn from arbitrary teacher context, while reducing forgetting by retaining initial performance on samples where the teacher has limited positive (even detrimental) impact. Moreover, our data partitioning can be achieved without any supervision.

Doing so, we see significant increases in the success rate (non-zero gains of the student) for all teacher-student pairings - from 32.5% with normal distillation to 92.5% with data partitioning. Our data-level regularization is the *only setting which allows for consistently positive transfer from weaker teachers, while retaining the transfer performance of normal distillation for much stronger teachers* and even outperforming normal distillation for equiperformant ones. In addition, it allows for the transfer of specialized knowledge (§5.1) and requires *no pairing-specific* hyperparameters. Unlike ensembling methods (Lakshminarayanan et al., 2017; Gontijo-Lopes et al., 2022; Sinha et al., 2021; Dietterich, 2000), our approach maintains original inference costs and handles high performance differences. Finally, we study architectural properties and their impact on the transfer process (§5.1) beyond the transfer method, and look into scalability to knowledge transfer from multiple models, where we find that simple sequential transfer can perform favorably when leveraging our transfer method, achieving clear improvements over transfer from just the single best teacher model.

Overall, our contributions can be summarized as: **(1)** We discover the consistent existence of complementary knowledge between arbitrary models pretrained on the same dataset - even if model families or performances differ. **(2)** We conduct extensive, exploratory studies to investigate the possibility of guaranteed model- and performance-independent transfer of the complementary knowledge without performance degradation. **(3)** We propose a successful transfer method motivated through the lens of continual learning, leveraging a confidence-based, hyperparameter-free data partitioning approach. **(4)** We provide studies on the relation of general model properties to general knowledge transfer, and **(5)** investigate knowledge transfer between multiple models. Code will be released upon acceptance.

## 2 RELATED WORK

Early works in knowledge distillation focus on compressing large teacher models into smaller student models. Bucila et al. (2006) achieve this by matching the soft targets of the teacher. Hinton et al. (2015) propose temperature scaling for lower probabilities. Recent works extend this with structural context: attention-transfer (Zagoruyko and Komodakis, 2017) encourages similar feature response

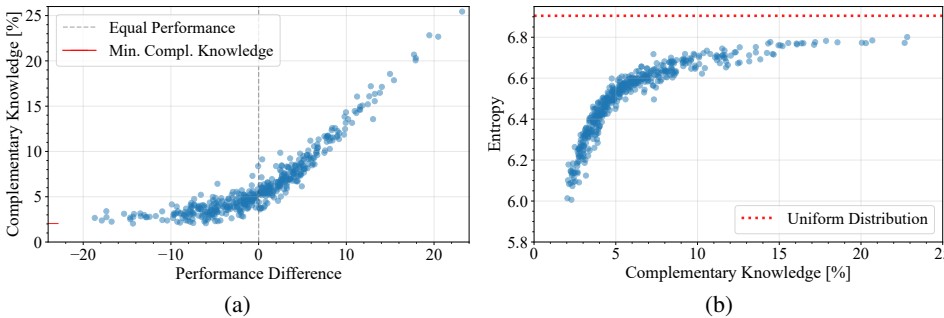

Figure 1: (a) We show the share of complementary knowledge ($\rho^{\text{pos}}$, perc. of pos. label flips of teacher w.r.t. student) against perform. differences $\Delta_{\text{acc}}$ for 466 teacher/student pairs, and find significant complementary context even for much weaker teachers. (b) Looking at the entropy of the compl. knowledge distribution over classes, we find context to be more specialized for weaker teachers.

patterns, Romero et al. (2015) and Zagoruyko and Komodakis (2017) propose (weighted) regression-guided student feature activations. However, these approaches are limited to matching student and teacher architectures. Tian et al. (2020) propose contrastive distillation, aligning teacher and student feature spaces with flexibility in feature dimensionalities, while highlighting performance overlap among most distillation objectives. These insights transfer to multiple teacher distillation (Luo et al., 2019; de Carvalho et al., 2022; Shen et al., 2019a;b), e.g. via teacher outputs reweighting (Wu et al., 2021; Liu et al., 2020; Yu et al., 2022; Yuan et al., 2020a). Unlike standard distillation, we look at knowledge transfer between arbitrary, **already trained models** - a much more difficult task, particularly when no restrictions (in contrast to e.g. (Yuan et al., 2020b)) are imposed *on relative performances or architectures, and initial knowledge should be retained*. On a conceptual level, this also connects to recent works on weak-to-strong model generalization for superalignment, for which our work can provide an orthogonal perspective and useful practical insights (Burns et al., 2023).

Wortsman et al. (2022a;b) show that when architecture and pretraining are *shared*, linear interpolation can be surprisingly effective for suitable loss basins (Neyshabur et al., 2020). Still, model selection through validation metrics is key for diverse collections. In contrast, we combine model knowledge *without restrictions* by leveraging perspectives from Continual Learning. Approaches are categorized into *regularization* (limit weight changes (Kirkpatrick et al., 2017; Schwarz et al., 2018; Li and Hoiem, 2016; Rebuffi et al., 2016; Castro et al., 2018)), *replay* (Rebuffi et al., 2016; Lopez-Paz and Ranzato, 2017; Chaudhry et al., 2019; Buzzega et al., 2020; Prabhu et al., 2020) through a data memory, and methods that restructure networks (Mallya and Lazebnik, 2018; Mallya et al., 2018; Zhang et al., 2020). On top of that, interpolation has also proven useful when continually adapting from a pretrained model (Stojanovski et al., 2022), which we include in our transfer study.

## 3 COMPLEMENTARY KNOWLEDGE BETWEEN PRETRAINED MODELS

Over recent years, thousands of models pretrained to completion on canonical datasets such as ImageNet have been made publicly available, with variations across all possible training choices (§1), which potentially impact generalization - the extent to which we wish to study in this section. In particular, we use `timm` (Wightman, 2019), comprising hundreds of models trained on ImageNet under varying conditions, and consider the image classification problem with input space $\mathcal{X}$ and label space $\mathcal{Y}$ with $c = |\mathcal{Y}|$ labels. Let $f(\cdot, \theta) : \mathcal{X} \to \mathbb{R}^c$ be a classifier with parameters $\theta \in \Theta$, logits $z = f(x, \theta)$ and softmax $\sigma(z)$ with $\sigma_j(z) = \exp(z_j)/\sum_i \exp(z_i)$ associated with samples $x \in \mathcal{X}$. We use $f_t$, $f_s$ to denote the pretrained teacher and student, with parameters $\theta_t$ and $\theta_s$ respectively. To evaluate the complementarity of knowledge between any $f_t$ and $f_s$, we follow the methodology of Lopes et al. (2022) and study the performance on the ImageNet validation set. Specifically, we measure *positive prediction flips* between $f_t$ and $f_s$, $\rho^{\text{pos}} = \frac{1}{n} \sum_{i=1}^{n} \rho_i^{\text{pos}}$, where $\rho_i^{\text{pos}}$ indicates a positive flip. This quantifies the proportion of samples correctly classified by the teacher but incorrectly by the student - the complementary knowledge. For the remainder of this work, we will refer to complementary knowledge and the existence of these positive flips interchangeably.

**Existence of complementary knowledge.** Using `timm`, we randomly select 466 ($f_t$, $f_s$) model pairs covering 301 unique models of varying architectures, sizes, and performances. In Fig. 1, we

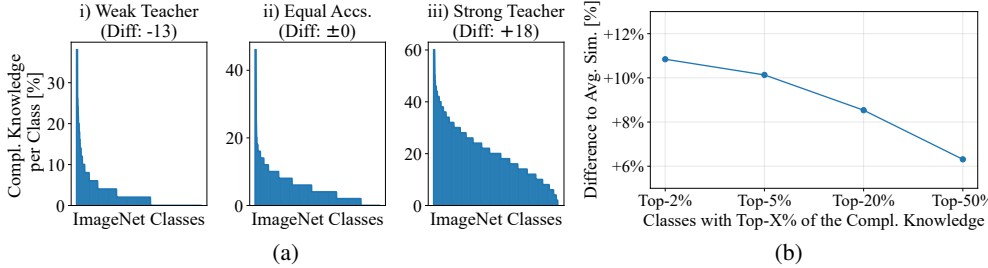

Figure 2: (a) *Sorted histograms of complementary knowledge per class* (positive prediction flips) as a share of total class samples for three teacher-student pairs (weak, equal and strong teacher). Complementary context is centralized around few classes. (b) *Semantic similarity* between top-X% classes sorted by complementary knowledge amount. Shown as relative difference to average class similarity. Classes with the most complementary knowledge are likely semantically related.

investigate the share of *complementary knowledge* against the difference in performance between $f_t$ and $f_s$. This is measured as the difference in validation accuracy $\Delta_{\text{acc}} = \text{acc}(f_t) - \text{acc}(f_s)$ for each pair. We find a large share of positive prediction flips when $f_t$ outperforms $f_s$. But even when $f_t$ notably underperforms the student model (up to 20%), a high share of positive flips can still be found. Converging to around 2%, this percentage is more than an order of magnitude higher than random noise - for $c = 1000$ classes, the probability of a model to correct a sample by chance is around 0.1% (e.g. randomized ResNet50 teachers show complementary knowledge of around only 0.03%). As such, our result indicate consistently existing complementary between any pretrained model.

**Understanding the complementary knowledge.** To figure out if the complementary knowledge is distributed evenly among classes or if it is systematically grouped within particular subsets, we analyze the distribution of prediction flips over all classes. For a selected subset in Fig. 2a, where classes are sorted by the amount of complementary knowledge encoded, we see some classes carrying a disproportionate amount of context, particularly in the case of a weaker teacher. For stronger ones, the complementary context the teacher can provide becomes more evenly distributed. This is further supported when looking at the entropy of these distributions for all $(f_t, f_s)$-pairs in Fig. 1 (right), where we see a clear trend towards more aggregated context for weaker teachers as the entropy goes down. In all cases however, a significant degree of context grouping remains. We denote these groups of classes with significant complementary knowledge as *relative areas of expertise*. This notion becomes even more evident as we investigate the semantic relation between them in Fig. 2b. For this, we measure the semantic similarity of classes containing the first 2%, 5%, 10%, 20% and 50% of positive flips (based on the per-class ranking by complementary knowledge as in Fig. 2a) using a pretrained language model (CLIP, Radford et al. (2021)). Comparing the measured similarity to the average similarity of all classes, we see a relative increase in semantic similarity by nearly double on average for the classes that encode the most complementary knowledge.

In summary, we observed that complementary knowledge between any pair of pretrained models exists, and that this knowledge which a pretrained teacher can pass to the student is centered around areas of expertise comprising semantically-related classes. The existence of ubiquitous complementary knowledge motivates our study of possible general-purpose knowledge transfer tools.

## 4  GENERAL KNOWLEDGE TRANSFER METHODOLOGY

This section explains possible knowledge transfer objectives, starting from standard knowledge distillation (§4.1 to our proposed extensions in §4.2, which highlights how and why this problem should be treated as a continual learning one. §4.3 extends this to multiple pretrained teachers.

### 4.1  KNOWLEDGE TRANSFER THROUGH KNOWLEDGE DISTILLATION

Knowledge Distillation (*KL distillation*) was pioneered by Hinton et al. (2015), which suggests minimizing the KL divergence between a teacher and a student model's soft targets $\sigma(z_t)$ and $\sigma(z_s)$:

$$\mathcal{L}_{KL} = {}^{T^2}\!/n \sum_{i=1}^{n} \text{KL}\left[\sigma(\mathbf{z}_{s,i}/T), \sigma(\mathbf{z}_{t,i}/T)\right], \tag{1}$$

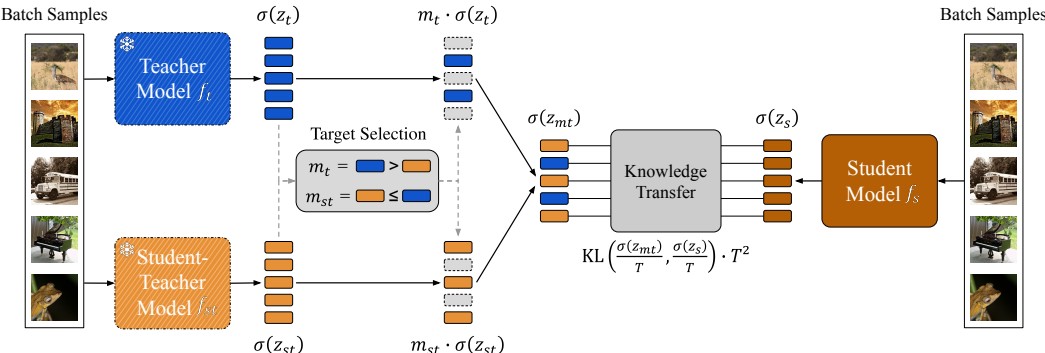

Figure 3: For general knowledge transfer between any pretrained models, we propose data-level regularization: Samples are separated based on if they should be taught through the teacher $f_t$ or retained via a frozen version of the initial student, $f_{st}$. All models forward the same batch, and outputs $\sigma(z_t)$ and $\sigma(z_{st})$ are merged on a sample-level via selection masks $m_t$ and $m_{st}$ derived from model confidences (Eq. 4). Lastly, we compute the KL-div. to the adapting student's ($f_s$) outputs $\sigma(z_s)$.

with temperature $T$. We use Eq. 1 as our base transfer objective (*KL-Dist transfer*), as it still remains popular and competitive (Beyer et al., 2022; Rajasegaran et al., 2020; Tian et al., 2020) (see Supp. for other objectives). KL distillation is often extended with auxiliary classification losses (e.g. cross-entropy $\mathcal{L}_{XE}$, (Hinton et al., 2015; Tian et al., 2020; Rajasegaran et al., 2020; Beyer et al., 2022)) to stabilize the distillation process. We denote the $\lambda$-weighted combination as *XE-KL distillation*, and the associated transfer as *XE-KL-Dist. transfer or XE-KL*:

$$\mathcal{L}_{dist} = \lambda \cdot \mathcal{L}_{KL} + (1 - \lambda) \cdot \mathcal{L}_{XE}. \tag{2}$$

Most knowledge distillation research considers the distillation from a trained teacher to an untrained student, in stark contrast to our goal of knowledge transfer between *pretrained* models while retaining student knowledge already gained *a priori*. And indeed, when applied to knowledge transfer between pretrained models in §5.1, standard knowledge distillation struggles to transfer knowledge without performance drops for most teacher-student pairs. We measure this via the *transfer delta* $\Delta_{transf} = acc(f_{s^{kt}}) - acc(f_s)$, which quantifies the change in the student's top-1 accuracy, with $f_{s^{kt}}$ being the student model following the knowledge transfer.

## 4.2 Knowledge Transfer Through Continual Knowledge Distillation

For successful knowledge transfer, a favorable trade-off between retaining existing student knowledge and incorporating complementary teacher knowledge is required. This bares semblance to Continual Learning (CL) frameworks (Kirkpatrick et al., 2016; Zenke et al., 2017; Aljundi et al., 2019), which aim to train models on incremental data streams without forgetting previously learned context.

**Constraining weight updates.** Unlike standard CL however, our problem is *not that of an incremental data stream*, but of continuous new learning signals from the teacher over the same transfer data. This excludes memory-based approaches, but permits regularization (see §2 for details), where Stojanovski et al. (2022) show that for continuous adaptation of trained models, momentum weight interpolation (MCL) proves more effective than existing regularizations. We thus extend our base XE-KL objective with MCL (*XE-KL-Dist+MCL transfer / XE-KL+MCL*). Weight interpolation in MCL retains a slow copy of the student weights $\theta_s^{slow}$ in addition to fast weights $\theta_s^{fast}$ updated during the transfer. At a predefined frequency $N$ and iteration $i$, $\theta_s^{slow}$ is updated via weight-space interpolation:

$$\theta_s^{slow,i+1} = \tau \cdot \theta_s^{slow,i} + (1 - \tau) \cdot \theta_s^{fast,i}, \tag{3}$$

with momentum $\tau$ guiding the weight constraint. Both $N$ and $\tau$ can be tuned to balance the plasticity-stability trade-off. However, as weight interpolation constrains weight updates for all samples equally, it struggles to leverage the relative areas of expertise of teachers to their full extent (c.f. §5.1).

**Constraining transfer data.** Instead of weight-level restrictions, we suggest regularization on the data-level by partitioning the transfer data into samples where the student can benefit from teacher feedback and ones where the prior knowledge should be retained. In particular, for samples essential to the transfer of complementary knowledge, we distill from the teacher $f_t$. However, for

the other sample set, we instead distill from the initial, frozen student model (denoted as $f_{st}$ for *student-teacher*), with the goal of retaining the initial model behavior. The selection of these data subsets follows a simple and effective greedy heuristic that assigns each sample depending on the highest prediction probability for the corresponding ground-truth class, giving data masks $m^{\text{t}}$ and $m^{\text{st}}$

$$m_i^{\text{t}} = \mathbb{I}\left[\sigma_j(\mathbf{z}_{t,i}) > \sigma_j(\mathbf{z}_{st,i})\right], \qquad m_i^{\text{st}} = \mathbb{I}\left[\sigma_j(\mathbf{z}_{t,i}) \leq \sigma_j(\mathbf{z}_{st,i})\right], \tag{4}$$

for each sample $i$ and $j = y_i$, and with $\sigma_j(z)$ the softmax output for class $j$. In practice, we find that the use of these masks provides enough stability to the transfer process *where auxiliary classification is no longer required*. In addition to that, we also find that this supervised heuristic can be replaced with a *fully unsupervised* one by assigning samples based on the maximum prediction probability for a sample, i.e. choosing the model *by confidence*. While this can suffer from overconfidence, in practice performance matches the supervised setting. This means that knowledge transfer can be performed *without access to labels*. We provide a visualization of our data partitioning (DP) in Figure 3. The final transfer approach which we refer to as *KL-Dist +DP transfer*, is thus given as:

$$\mathcal{L}_{\text{dist}} = T^2/n \sum_{i=0}^{n} m_i^{\text{t}} \cdot \text{KL}\left[\sigma(\mathbf{z}_{s,i}/T), \sigma(\mathbf{z}_{t,i}/T)\right] + m_i^{\text{st}} \cdot \text{KL}\left[\sigma(\mathbf{z}_{s,i}/T), \sigma(\mathbf{z}_{st,i}/T)\right]. \tag{5}$$

As can be seen, *KL-Dist + DP transfer* (or *KL+DP*) requires no additional hyperparameters compared to standard knowledge distillation, with strong robustness towards temperature choices (c.f. Supp.).

## 4.3 MULTI-TEACHER KNOWLEDGE TRANSFER

With multiple models available in model zoos, studying how to transfer context between multiple experts is a natural extension, for which we suggest studying three approaches. Firstly, in line with standard multi-teacher knowledge distillation, all teachers can be used at once for knowledge transfer (*parallel*), while still leveraging our proposed transfer regularizer described above to ensure positive transfer. In particular, the greedy data selection is extended to produce data subsets *for each* respective teacher model. Secondly, the multi-model transfer process can be done *sequentially* in line with the continual treatment of the single-model transfer process. After each teacher model transfer, the distilled student is treated as the (new) pretrained student for the subsequent transfer step. Finally, our experimental section also investigates the use of Model Soups (Wortsman et al., 2022a) for the problem of architecture-independent knowledge transfer. Here, the student model is independently distilled from each teacher model, producing a set of distilled model variants $\{f_s^i\}_{i \in 1,...,K_t}$ with $K_t$ teacher models. After transfer, simple weight interpolation between all variants is performed (*soup*).

## 5 EXPERIMENTAL STUDY ON EFFECTIVE KNOWLEDGE TRANSFER

We first conduct a large-scale study of knowledge transfer approaches (c.f. §4.1-4.3) in §5.1, highlighting the advantage of a continual learning approach, and particularly the superiority of our data-partitioning method. For exploration, we use a supervised variant (Eq. 4) but show in Tab. 3 that the unsupervised variant matches the performance. Finally, we investigate the relation of model properties and general transfer success (also §5.1), and study transfer from multiple models in §5.2. **Experimental details.** We use NVIDIA 2080Ti compute clusters with PyTorch 1.13.1 (Paszke et al., 2017) and ffcv 0.0.3 (Leclerc et al., 2022) for fast data-loading. While performance may be slightly impacted, relative changes are retained (Leclerc et al., 2022), allowing large-scale experiments on reasonable compute. For our large-scale evaluations of transfer approaches, we use a 10% stratified ImageNet subset (subsampling per class), then validate our main claims on the full ImageNet dataset. We perform transfer over a constrained budget of 20 epochs to study methods for general-purpose knowledge transfer with practical requirements. Other optimization parameters are determined via grid searches (see Supp.). **Code:** github.com/ExplainableML/General-Knowledge-Transfer.

### 5.1 EVALUATION OF DIFFERENT APPROACHES FOR KNOWLEDGE TRANSFER

**Effectiveness of standard knowledge distillation for knowledge transfer.** To study the suitability of standard KL distillation for general-purpose knowledge transfer, we select 400 teacher-student pairs (Tab. 5 in Supp. for details), all of which exhibit a significant percentage of complementary knowledge (c.f. §3). Across these pairs for each student model, we measure the percentage of teachers for which a positive transfer delta $\Delta_{\text{transf.}}$ is obtained. Results are visualized in Fig. 4a, and reveal that for the majority of students there are less than $40\%$ of teachers with performance increases. An

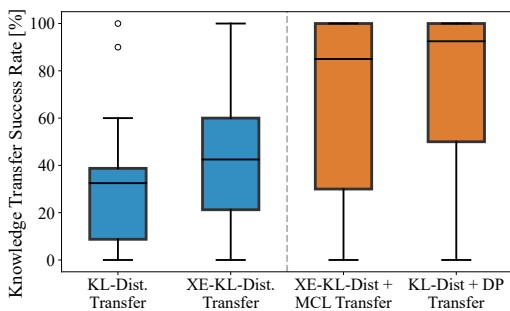 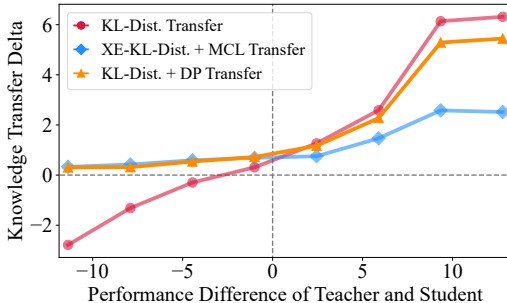

(a) Percentage of successful transfers.      (b) Correlation teacher properties & transfer success.

Figure 4: **(a)** Share of teachers resulting in positive knowledge transfer (success rate) for knowledge distillation variants (blue) and continual learning extensions (orange). Each box represents 400 transfer experiments, with a clear increase for continual learning setups. **(b)** Transfer delta by binned teacher-student performance difference. For more robust reporting, we show the mean transfer delta of the top 25% for each bin/approach with the same 400 teacher-student pairs. The results show *KL-Dist+DP Transfer* enabling consistent gains from weaker **and** stronger teachers.

Table 1: Knowledge transfer results on full ImageNet, from four teacher to eight selected student models (see Supp.Tab. 6). The results provide further evidence that a continual learning approach to the transfer problem allows for more consistent knowledge transfer across arbitrary model pairs.

| Students | Type | Acc. | # Param. | $\Delta_{transf.}$ KL-Dist. | $\Delta_{transf.}$ KL-Dist+DP |
|---|---|---|---|---|---|
| XCiT-P16 (El-Nouby et al., 2021) | Trafo | 82.89 | 189.10 | 0.13 ($\pm$1.01) | **0.65** ($\pm$0.26) |
| PiT-B (Heo et al., 2021) | Trafo | 82.44 | 73.76 | 0.33 ($\pm$0.86) | **0.55** ($\pm$0.25) |
| PiT-XS (Heo et al., 2021) | Trafo | 78.19 | 10.62 | 0.10 ($\pm$0.49) | **0.40** ($\pm$0.12) |
| SeNet154 (He et al., 2018) | CNN | 81.23 | 115.09 | -0.05 ($\pm$0.26) | **0.27** ($\pm$0.17) |
| ConvNext (Liu et al., 2022) | CNN | 84.57 | 50.22 | -0.51 ($\pm$0.85) | **0.33** ($\pm$0.14) |
| ResNetV2 (He et al., 2016b) | CNN | 82.80 | 25.55 | -0.09 ($\pm$0.34) | **0.23** ($\pm$0.08) |
| Mixer-B16 (Tolstikhin et al., 2021) | MLP | 82.30 | 59.88 | -0.29 ($\pm$0.58) | **0.15** ($\pm$0.13) |
| ResMLP-24 (Touvron et al., 2021) | MLP | 80.76 | 30.02 | 0.15 ($\pm$0.36) | **0.33** ($\pm$0.19) |

additional classification loss (*XE-KL*, §4.1) can raise the median success rate slightly above 40%. In both cases, however, overwriting pre-existent knowledge more often than not overshadows the benefits gained from actual knowledge transfer, particularly when transferring from a weaker teacher model as shown in Fig. 4b (*KL-Dist. Transfer*), where absolute performance changes after transfer are visualized against initial teacher-student performance differences (as measured on the separate validation data). In addition, we find that these limits also hold when deploying simple additional regularizers such as label smoothing (54%, Yuan et al. (2020b)), with consistently negative transfer for weaker teachers, and reduced effectiveness for stronger ones.

**Leveraging continual learning.** Treating general knowledge transfer as a continual learning task through weight regularization (*XE-KL+MCL*) raises median success rates significantly (80%, Fig. 4a). However, we find a lack of efficacy when knowledge is specialized to areas of expertise, and when teachers are stronger, which we address with data-level regularization (*KL+DP*), raising success rates to 92.5%. As shown in Fig. 4b, these gains can be attributed to *positive transfer deltas even for much weaker teachers* (see performance differences much lower than zero), and, unlike strict weight-level regularization in e.g. MCL, *barely limit gains from much stronger teachers*. Indeed, we find that for a number of stronger teachers, particularly where performance differences are not as striking, data-level regularization can even offer an edge over normal distillation.

**Full ImageNet experiments.** We extend our experiments from above to full ImageNet in Tab. 1, and verify that insights transfer, with *KL+DP* performing significantly more reliably than normal distillation. We also provide a more detailed overview in Table 2, highlighting successful transfer from both stronger, *but particularly weaker teachers as well* - even if student models are already strong ($\approx$ +0.3% for PiT-B (Heo et al., 2021), 82.44% ImageNet top-1 and ConvNeXt (Liu et al., 2022), 84.57%, with a nearly 5% performance differential). For additional results on ImageNet and model information, we refer to tables 6 and 7 in the supplementary. In addition, the supplementary also reveals similar transfer success using *KL+DP* for other datasets such as CUB200 (Wah et al.,

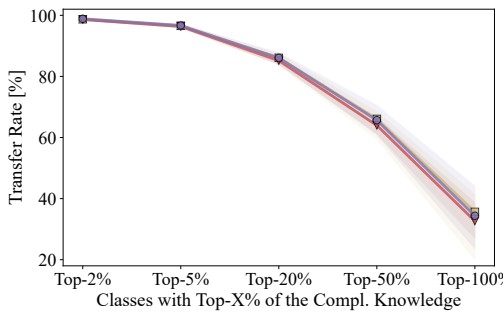
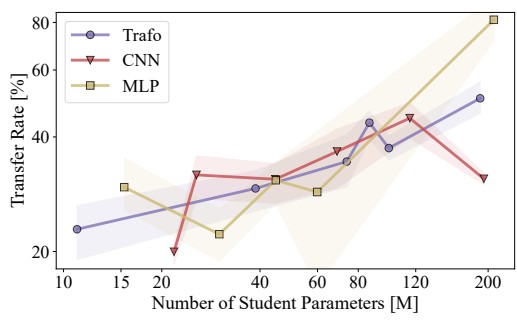

(a) Transfer rates for the top-X% pos. flip classes.     (b) Transfer rate by student size and model family.

Figure 5: **(a)** Transfer rates for the sets of classes containing the top 2% - 100% complementary knowledge per model family. **(b)** Transfer rates versus student size separated by model family.

2011), StanfordCars (Krause et al., 2013), and Caltech256 (Griffin et al., 2007). Furthermore, we leverage full ImageNet runs in Tab. 3 to demonstrate that our *unsupervised* variant of KL+DP transfer (§4.2) performs comparably and in some cases even better than its supervised counterpart (e.g. $\Delta_{transf.} = 0.55$ vs $\Delta_{transf.} = 0.95$ for PiT-B students). *This showcases that general knowledge transfer can be done even without supervision by leveraging data-level continual learning regularization.*

We do find increased variance due to the unsupervised data selection. To better understand it, we study the number of positive (teacher correct, student incorrect) and negative flip samples (vice versa) that are assigned to the teacher. Ideally, this only includes positive flip samples. For model pairs presented in Table 6 using *KL-Dist. + DP transfer*, we find that 72% of positive and 9% of negative flip samples are assigned to the teacher. This means that while simple confidence-based partitioning does not perfectly assign samples, it still strongly aligns with respective areas of expertise.

Overall, we find very promising transfer across teacher-students pairs - even without supervision. While gains fall short of the total complementary knowledge §3 - attributable to the trade-off between retention and transfer - we believe our results *to offer a strong proof-of-concept for future research and the potential of truly general knowledge transfer*.

**Complementary knowledge drives transfer gains.** Our experiments above show clear gains

Table 2: *Examples for the transfer deltas* of two student models each distilled with a stronger and weaker teacher model on full ImageNet using our KL-Dist+DP transfer approach.

| Teacher → | Volo-D2 | | ResMLP-36 | |
| Student ↓ | $\Delta_{acc}$ | $\Delta_{transf.}$ | $\Delta_{acc}$ | $\Delta_{transf.}$ |
|---|---|---|---|---|
| PiT-B | +2.75 | 0.86 | -2.67 | 0.31 |
| ConvNext | +0.62 | 0.44 | -4.80 | 0.26 |

Table 3: *Comparison between supervised and unsupervised KL-Dist+DP transfer on ImageNet* for eight selected students and four teachers, respectively. Results show that fully unsupervised knowledge transfer between experts is not only possible but can even outperform supervised transfer.

| Students | $\Delta_{transf.}$ KL+DP Supervised | $\Delta_{transf.}$ KL+DP Unsupervised |
|---|---|---|
| XCiT-P16 (El-Nouby et al., 2021) | 0.65 (±0.26) | **0.96** (±0.42) |
| PiT-B (Heo et al., 2021) | 0.55 (±0.25) | **0.95** (±0.52) |
| PiT-XS (Heo et al., 2021) | **0.40** (±0.12) | 0.35 (±0.18) |
| SeNet154 (He et al., 2018) | **0.27** (±0.17) | 0.25 (±0.15) |
| ConvNext (Liu et al., 2022) | **0.33** (±0.14) | 0.28 (±0.14) |
| ResNetV2 (He et al., 2016b) | **0.23** (±0.08) | **0.23** (±0.09) |
| Mixer-B16 (Tolstikhin et al., 2021) | 0.15 (±0.13) | **0.17** (±0.15) |
| ResMLP-24 (Touvron et al., 2021) | **0.33** (±0.19) | 0.29 (±0.24) |

when conducting *KL+DP* transfer across all kinds of model pairs. In this part, we show that gains indeed stem from the transfer of the complementary knowledge between these pairs (c.f. §3). For that, we analyze the student's improvement per class relative to the available complementary knowledge per class, denoted as *transfer rate*, for all 400 transfer runs studied in Fig.4a. Results in Fig. 5a plot the transfer rate against classes containing the top-$X\%$ of complementary knowledge. After sorting classes by the share of complementary knowledge (c.f. in Fig. 2a, we look at how the first $X\%$ of complementary knowledge are transferred). Following our notation from §5.1, smaller percentages mostly contain flips from a teacher's relative area of expertise, which gets softened for larger values. Indeed, when utilizing our proposed *KL+DP* transfer, we find a clear indication where complementary knowledge associated with stronger areas of expertise in the teacher model have a near-guaranteed chance to be transferred to the student model. This drops when moving towards flips associated with less represented classes. Note that without preference towards relative areas of expertise, one would expect a horizontal line at the average transfer rate. This shows that our CL-based treatment of knowledge transfer allows any teacher model to impart specific knowledge and that one can *explicitly guide the context learned by a student model by choosing a teacher with a suitable area of expertise*.

Table 4: *Transfer from multiple teachers* in sequential, parallel and soup-based fashion (c.f. §4.3). We find sequential transfer to perform favorably.

| Students | Type | Acc. | # Param. | $\Delta_{transf.}$ Single Teacher | | | $\Delta_{transf.}$ Multiple Teachers | | |
|---|---|---|---|---|---|---|---|---|---|
| | | | | Mean | Min | Max | Sequ. | Parallel | Soup |
| XCiT-P16 (El-Nouby et al., 2021) | Trafo | 82.89 | 189.1 | 0.73 | 0.36 | 0.93 | **0.97** | 0.91 | 0.63 |
| Twins (Chu et al., 2021) | Trafo | 83.68 | 99.27 | 0.22 | 0.20 | 0.26 | **0.65** | 0.30 | 0.21 |
| PiT-B (Heo et al., 2021) | Trafo | 82.44 | 73.76 | 0.64 | 0.31 | 0.86 | **1.04** | 0.77 | 0.69 |

**Properties of a good student model.** We conclude our single model transfer studies by investigating if and how specific student model properties can facilitate the reception of complementary knowledge. Across our experiments performed in Sec. 5.1, we find in Fig. 5b that the transfer rate of complementary knowledge has *a significant relationship with the model capacity* (parameter count). We observe this general trend across architectural families irrespective of the visual inductive biases encoded. However, we highlight that while larger models are generally more receptive towards additional context, CNN-style architecture, or generally *models with stronger visual inductive biases, are more prone to have their existing knowledge overwritten*, resulting in a lower distillation delta. See Supp. for additional details and experiments.

## 5.2 KNOWLEDGE TRANSFER FROM MULTIPLE PRETRAINED MODELS

Finally, we study transfer from multiple teachers on full ImageNet, testing sequential and parallel *KL+DP* transfer, and a model soups variant by interpolating between transferred student variants (c.f. §4.3). Our experiments include three students and teachers (Supp. B.6), where we focus on Transformers, which in our experiments have shown the highest aptitude towards knowledge reception. Results are shown in Tab. 4, where we compare to the performance of single-teacher transfer.

We find that students can consistently gain knowledge from each teacher when transferring sequentially. However, as the student improves, returns diminish until the transfer deltas plateau, as forgetting becomes more prevalent as we move further from the initial student pretraining. Nevertheless, the sequential transfer of three teachers achieves an average gain of 59% compared to the best single teacher transfer delta (e.g. $\Delta_{transf} = 0.26 \rightarrow \Delta_{transf} = 0.65$ for Twins (Chu et al., 2021) or $\Delta_{transf} = 0.86 \rightarrow \Delta_{transf} = 1.04$ for PiT-B (Heo et al., 2021)). At the opposite end, we find vanilla KL-Dist. transfer to be unsuccesful in the multi-teacher setting, underlining the benefits of *KL+DP* transfer (see also Supp.). Furthermore, while we found consistent knowledge gain irrespective of the teacher order, our experiments indicate that a descending teacher sequence (i.e. strongest first) can actually incur disproportionately higher forgetting, as the model moves away from its base knowledge more quickly. Finally, unlike sequential transfer, parallel transfer of multiple teachers does not improve over the best single teacher transfer performance. This is due to a reduced amount of retention happening as the respective subset regularization (c.f. §4.3) does not retain enough samples for active knowledge retention. Finally, we find weight averaging of student variants distilled with each respective teacher to perform worst (underperforming the best single-teacher transfer), which we attribute to a lack of interpolatability and a subsequent reduction in knowledge retention.

## 6 CONCLUSION

In this work, we show that **any** pairing of models trained on the same dataset with different training protocols (e.g. changes in architecture or optimization) exhibits significant complementary knowledge - data context encoded in one model and not the other. Based on the presence of complementary knowledge, we offer a first exploration into a general mechanism to transfer it between any pair of models without performance degradation and reliance on external ranking measures. This unlocks any model repository as a resource for model gains, and the option to improve large models with context from weaker, lower resource ones. Our large-scale experiments reveal the limits of simple knowledge distillation as a general transfer mechanism, and suggest extensions through the lens of continual learning and confidence-based data partitioning. This raises the transfer success rate from under $40\%$ to over $92\%$, with positive transfer from both stronger and weaker teachers. We also provide insights into general model properties that facilitate transfer, finding model capacity and reduced visual inductive biases to be beneficial. Finally, we showcase transfer from multiple models with our transfer mechanism. Overall, we provide the experimental motivation and first steps towards general-purpose complementary knowledge transfer tools between arbitrary model architectures.

## ACKNOWLEDGEMENTS

Karsten Roth thanks the European Laboratory for Learning and Intelligent Systems (ELLIS) PhD program and the International Max Planck Research School for Intelligent Systems (IMPRS-IS) for support. This work was supported by DFG project number 276693517, by BMBF FKZ: 01IS18039A, by the ERC (853489 - DEXIM), by EXC number 2064/1 – project number 390727645.

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

APPENDIX

# FANTASTIC GAINS AND WHERE TO FIND THEM: ON THE EXISTENCE AND PROSPECT OF GENERAL KNOWLEDGE TRANSFER BETWEEN ANY PRETRAINED MODEL

## A   IMPLEMENTATION DETAILS AND EXPERIMENTAL INSIGHTS

In this section, we describe the implementation details of our experiments to evaluate the effectiveness of different approaches and techniques for transferring complementary knowledge between pretrained expert models trained on the same dataset.

For our initial and exploratory experiments we use a 10% stratified subset of ImageNet (Deng et al., 2009) to reduce runtimes, in order to conduct a wider range of experiments across a large number of model pairs. In detail, we drew 130 samples per class using the standard ImageNet validation set for evaluation. All of our experiments utilize an SGD optimizer with momentum 0.9 and weight decay 1e-3. Further hyperparameters were individually tuned for each investigated transfer approach.

### A.1   IMPLEMENTATION OF DISTILLATION-BASED KNOWLEDGE TRANSFER VARIANTS

To set the learning rate for our default knowledge distillation based transfer approach using KL divergence (as in Section 4.1), we conducted a parameter search over a set of 33 teacher-student pairs randomly selected from the `timm` Wightman (2019) library, with learning rates lr $\in \{$1e-2, 1e-3, 1e-4, 1e-5$\}$, for which we found a learning rate of 1e-4 to generally work best, albeit regardless of chosen values, the average transfer delta was consistently negative.

Following Section 4.1, we also extend KL distillation with a cross-entropy classification loss. In this case, hyperparameters were determined over a grid comprising learning rates lr $\in \{$1e-2, 1e-3, 1e-4, 1e-5$\}$, softmax temperatures $T \in \{0.1, 1, 4, 10\}$ and weightings $\lambda \in \{0.05, 0.1, 0.25, 0.5\}$. Again, we found that a learning rate of 1e-4 was the most effective on average, but found particular variance in the weighting $\lambda$, where we observed that a larger $\lambda$ value - placing higher emphasis on distillation - is better suited for transferring knowledge from a stronger teacher to a weaker student, while a smaller $\lambda$ seems to be preferable when transferring knowledge from a weaker teacher to a stronger student. This further highlights the trade-off between knowledge gain and retention, where for a weaker teacher, retention plays a much more crucial part to ensure overall high performance, as student knowledge is overwritten.

For the softmax temperature, we found that a small temperature of 0.1 limits the decrease in the student's performance when transfering from a weaker teacher model, but also limiting the knowledge transfer in general. This results in only small increases in the student's performance even when transferring from a stronger teacher model. Hinton et al. (2015) propose to use a larger temperature of 4 to match soft targets to better represent smaller probabilities in the output of a single sample. However, we do not find larger temperatures to benefit the transfer performance.

In general, we find that particularly the temperature and weighting parameter guide the aggressiveness of the distillation-based transfer approach, which is highly dependent on the observed teacher and student dynamic of the provided pair of pretrained expert models. The high variance across such arbitrary model pairs makes normal knowledge distillation, even paired with an additional classification loss for stability, *not well suited as a general knowledge transfer tool*.

### A.2   IMPLEMENTATION OF CONTRASTIVE DISTILLATION KNOWLEDGE TRANSFER

While knowledge distillation approaches matching the soft targets of the teacher and student model remain popular, various recent approaches argue that more structural knowledge can be transferred by encouraging the student model to also match intermediate representations of the teacher model (Liu et al., 2019; 2020; Wu et al., 2021; Park and No, 2021). Thus, in this section, we highlight results of our exploration on the feasibility of using intermediate representations and their relations to transfer knowledge between pretrained experts.

We particularly follow Tian et al. (2020), who propose to extend the basic knowledge distillation approach of Hinton et al. (2015) by aligning the feature representations of the teacher and the student models. Here, the student is encouraged to provide feature representations close to the ones of the teacher for similar images while repelling the feature representation of dissimilar images. Unlike other existing distillation approaches operating on feature representations, such a contrastive approach puts less restrictions on the architectures of the teacher and the student model, particularly because the feature representations of both models can be cheaply projected into a common feature space using a learned projection layer for both models. This enables the distillation between models of different architectures, and allows us to explore an alternative to our utilized base KL Distillation objective for general knowledge transfer (Sec. 4.1).

To assess the feasibility of representation-matching for knowledge transfer between expert models, we implement two contrastive learning approaches. First, we utilize a simple approach that encourages the distances between the feature representations of a pair of images to be similar for both the teacher and the student model. Hence, if two images result in similar feature representations in the teacher's embedding space, the student is encouraged to also provide feature representations with close proximity in their respective embedding space. Such relative similarity-based matching has seen success in standard supervised contrastive learning, such as in (Roth et al., 2021; 2022). Using $t$ and $s$ to denote teacher and student respectively, this gives

$$\mathcal{L}_{\text{CD}} = \text{KL}\left(\sigma(S_s), \sigma(S_t)\right), \tag{6}$$

where $S$ is a similarity matrix containing the cosine similarities of the normalized feature representations of the current batch ($S_{ij} = \cos \text{sim}(\text{norm}(s_i), \text{norm}(s_j)), \forall i, j \in 0, ..., n$). We denote this approach as *CD Distillation*.

Secondly, we implement the contrastive representation distillation approach (*CRD distillation*) of Tian et al. (2020). As noted, CRD distillation directly aligns representations by encouraging the student to be close to the teacher for positive pairs (different augmentations of the same image) while pushing apart feature representations of negative pairs (images of different classes). The respective objective is thus given as:

$$\mathcal{L}_{\text{CRD}} = \arg\max_{f_s} \max_{h} \mathbb{E}_{q(t,s|C=1)}[\log h(t,s)] + k\mathbb{E}_{q(t,s|C=1)}[\log(1 - h(t,s))], \tag{7}$$

where we utilize $t, s$ as shorthand for respective teacher and student representations. In addition, we use $h : t, s \rightarrow [0, 1]$ to represent a discriminator estimating whether the feature representation $t$ and $s$ are drawn from the same joint distribution or from the respective marginal product. In this setup, $k$ denotes the number of negative pairs drawn from the product of marginals.

Both contrastive distillation approaches compute the overall distillation loss $\mathcal{L}_{\text{dist}}$ as a weighted combination of the respective contrastive loss $\mathcal{L}_{\text{CD}}$ or $\mathcal{L}_{\text{CRD}}$ and a cross-entropy classification loss $\mathcal{L}_{\text{XE}}$ as also used in standard KL Divergence distillation objectives Beyer et al. (2022); Rajasegaran et al. (2020).

For CD distillation based knowledge transfer, we tested different weightings between the contrastive loss and the classification loss as well as different learning rates on a small set of teacher-student combinations. On a similar hyperparameter grid as noted in the previous section, we found an equal weighting of both losses in combination with a learning rate of 1e-4 to be most suitable on average, thought with a similar trade-off as depicted in Section A.1. For the CRD distillation transfer, we found the hyperparameters as provided in Tian et al. (2020) to work well.

### A.3 IMPLEMENTATION OF CONTINUAL LEARNING BASED TRANSFER APPROACHES

Finally, we describe hyperparameters and the corresponding hyperparameter studies utilized for our continual learning extension to distillation-based knowledge transfer (see Section 4.2), in particular the setup for *XE-KL-Dist+MCL transfer* and *KL-Dist+DP transfer*.

For *XE-KL+MCL transfer*, we conducted a parameter search on a learning rate grid with the same resolution as before. However, as there are several other parameters to validate, we only test lr $\in \{$1e-2, 1e-3$\}$. In addition to that, we follow Stojanovski et al. (2022) and test the momentum for values in $\tau \in \{0.99, 0.999, 0.9999\}$) and the interpolation frequency $N \in \{2, 10, 50, 100\}$). For the weighting against the classification objective, $\lambda$, we test 0.5 and 0.7. We conducted the

Table 5: Selection of student and teacher models used for the experiments on the 10% ImageNet subset. Each set of models was selected to contain multiple architecture types and cover a wide range of model sizes and performance levels.

| Student Models | Type | Acc. | # Param. |
|---|---|---|---|
| XCiT-Large-24-P16 El-Nouby et al. (2021) | Trafo | 82.89 | 189.10 |
| ViT-Base-P16 Dosovitskiy et al. (2021) | Trafo | 84.53 | 86.57 |
| PiT-B Heo et al. (2021) | Trafo | 82.44 | 73.76 |
| ViT-Relpos-Medium-P16 | Trafo | 82.46 | 38.75 |
| PiT-XS Heo et al. (2021) | Trafo | 78.19 | 11.00 |
| PiT-XE-dist Heo et al. (2021) | Trafo | 79.31 | 11.00 |
| IG-ResNext101-32x16d Xie et al. (2017) | CNN | 84.17 | 194.03 |
| Gluon-SeNet154 He et al. (2018) | CNN | 81.23 | 115.09 |
| Wide-ResNet50-2 He et al. (2016a) | CNN | 81.46 | 68.88 |
| ResNet101 He et al. (2016a) | CNN | 79.54 | 44.57 |
| ResNetV2-50 He et al. (2016b) | CNN | 80.40 | 25.55 |
| ResNet34-v1b He et al. (2016a) | CNN | 74.59 | 21.80 |
| ResNetv2-50-dist He et al. (2016b) | CNN | 82.80 | 25.55 |
| Mixer-L16 Tolstikhin et al. (2021) | MLP | 72.07 | 208.20 |
| Mixer-B16-miil Tolstikhin et al. (2021) | MLP | 82.30 | 59.88 |
| Mixer-B16 Tolstikhin et al. (2021) | MLP | 76.61 | 59.88 |
| ResMLP-36 Touvron et al. (2021) | MLP | 79.77 | 44.69 |
| ResMLP-24 Touvron et al. (2021) | MLP | 79.39 | 30.02 |
| ResMLP-12 Touvron et al. (2021) | MLP | 76.66 | 15.35 |
| ResMLP-24-dist Touvron et al. (2021) | MLP | 80.76 | 30.02 |

| Teacher Models | Type | Acc. | # Param. |
|---|---|---|---|
| ConvNext Liu et al. (2022) | CNN | 86.64 | 197.77 |
| VOLO-D4 Yuan et al. (2021) | Trafo | 85.88 | 192.96 |
| RegNety-320 Radosavovic et al. (2020) | CNN | 80.80 | 145.05 |
| VGG13 Simonyan and Zisserman (2015) | CNN | 71.60 | 133.05 |
| RegNetx-320 Radosavovic et al. (2020) | CNN | 80.24 | 107.81 |
| TWINS Chu et al. (2021) | Trafo | 83.68 | 99.27 |
| SWSL-ResNext101 Xie et al. (2017) | CNN | 84.29 | 88.79 |
| SWIN-S3 Liu et al. (2021b) | Trafo | 83.93 | 71.13 |
| TWINS-pcpvt Chu et al. (2021) | Trafo | 83.14 | 60.99 |
| VOLO-D2 Yuan et al. (2021) | Trafo | 85.19 | 58.68 |
| ResMLP-36 Touvron et al. (2021) | MLP | 79.77 | 44.69 |
| DLA102 Yu et al. (2018) | CNN | 78.03 | 33.27 |
| SWSL-ResNext50 Xie et al. (2021) | CNN | 82.18 | 25.03 |
| ViT-P16 Dosovitskiy et al. (2021) | Trafo | 81.40 | 22.05 |
| gMLP-S16 Liu et al. (2021a) | MLP | 79.64 | 19.42 |
| COAT-lite Xu et al. (2021) | Trafo | 79.09 | 11.01 |
| MixNet Tan and Chen (2019) | MLP | 78.98 | 7.33 |
| RegNety-006 Radosavovic et al. (2020) | CNN | 75.25 | 6.06 |
| MixNet Tan and Chen (2019) | MLP | 76.00 | 4.13 |
| XCiT-nano-12-P8 El-Nouby et al. (2021) | Trafo | 73.92 | 3.05 |

parameter search as a random search over the parameter grid. Ultimately, we found a parameter setting using a high momentum of 0.9999 in combination with a high interpolation frequency (every other iteration) and a learning rate of 0.01 with weight score 0.7 to work best on average. Unlike simple KL Distillation based transfer, a fixed hyperparameter combination now results in both a positive transfer delta on average, and a significantly increased number of teachers from which each student can learn from (c.f. Fig. 4a)

For our final proposed *KL+DP transfer* approach, we again conducted a similar parameter search. However, unlike *XE-KL+MCL transfer*, the *KL+DP* approach **does not introduce additional hyperparameters** compared to the standard KL distillation based setup. Consequently, we utilize a grid of $lr \in \{1e\text{-}3, 1e\text{-}4\}$, $\lambda \in \{0.5, 0.75, 0.9, 1\}$ and $T \in \{0.1, 1, 10\}$. Note that while we ablated the use of an external cross-entropy classification loss, we found the best performance to consistently come for $\lambda = 1$ - by turning of the auxiliary classification objective. This provides strong evidence that an external measures for training stability are no longer required. Finally, across all remaining experiments, we utilize a learning rate of 1e-4 and a temperature of 1. While more in-depth parameter searches could likely provide a parameter combination that would improve the average success rate, we believe that results achieved in its current setting to offer sufficient *proof-of-concept*.

## A.4 MODEL LISTS: LARGE-SCALE STUDIES ON STRATIFIED IMAGENET SUBSETS

Table 5 presents a comprehensive summary of the pretrained teacher and student models employed in our evaluation of various transfer techniques on the 10% subset of the ImageNet dataset (§5.1). These models were carefully chosen to encompass diverse architecture families, demonstrate varying performance levels, and exhibit a range of model sizes. This selection allows us to thoroughly examine the efficacy of knowledge transfer methods in different scenarios and settings. Note that for the exploration of complementary context (§3) we leveraged an even broader set of 466 teacher-student pairs comprising of 301 individual pretrained models randomly drawn from the `timm` Wightman (2019) library.

## A.5 MODELS EVALUATED ON FULL IMAGENET

Table 6 showcases the detailed specifications of the student and teacher models employed in our full-scale ImageNet experiments (refer to Section 5.1). In the context of knowledge transfer from multiple teacher models (§4.3), we utilized the same set of teacher models in combination with a subset of student models.

Table 6: Selection of student an teacher models used for the experiments on full ImageNet. The student models were selected to contain multiple architecture types and cover a wide range of model sizes and performance levels.

| Student Models | Type | Acc. | # Param. |
|---|---|---|---|
| XCiT-large-24-p16 El-Nouby et al. (2021) | Trafo | 82.89 | 189.10 |
| PiT-B Heo et al. (2021) | Trafo | 82.44 | 73.76 |
| PiT-XS Heo et al. (2021) | Trafo | 78.19 | 11.00 |
| Gluon-SeNet154 He et al. (2018) | CNN | 81.23 | 115.09 |
| ConvNext Liu et al. (2022) | CNN | 84.57 | 50.22 |
| ResNetV2-50-dist He et al. (2016b) | CNN | 82.80 | 25.55 |
| Mixer-B16-miil Tolstikhin et al. (2021) | MLP | 82.30 | 59.88 |
| ResMLP-24-dist Touvron et al. (2021) | MLP | 80.76 | 30.02 |

| Teacher Models | Type | Acc. | # Param. |
|---|---|---|---|
| SWSL-ResNext101 Xie et al. (2017) | CNN | 84.29 | 88.79 |
| VOLO-D2 Yuan et al. (2021) | Trafo | 85.19 | 58.68 |
| ResMLP-36 Touvron et al. (2021) | MLP | 79.77 | 44.69 |
| CoaT-lite-mini Xu et al. (2021) | Trafo | 79.09 | 11.01 |

# B EXTENDED EXPERIMENTAL RESULTS

In this section, we present additional experimental results of our experiments conducted in Section 5.

## B.1 ADDITIONAL EXPERIMENTS ON VARIANTS OF DISTILLATION-BASED KNOWLEDGE TRANSFER

In the following subsection, we present supplementary experiments conducted to enhance the performance of knowledge transfer variants for knowledge transfer among pretrained models.

**Using a cross-entropy plus distillation transfer objective.** As an alternative to the KL divergence used in Equation (1) we additionally investigated the potential of using a cross-entropy loss between the soft targets of the teacher and the student model, similar to Hinton et al. (2015). However, our results showed no advantage in using a cross-entropy loss over KL divergence. In fact, we observed an average transfer delta that was 1.2 percentage points lower when using cross-entropy loss compared to KL divergence on a set of 60 teacher-student pairs. We also explored the use of a warmup epoch where only the student model's linear layers are trained using KL divergence loss, but found no improvement in transfer performance.

**Restricting the set of classes for computing the distillation-based transfer loss.** In our supplementary experiments, we investigate the impact of limiting the distillation loss to focus only on the top-10 or top-100 most probable classes. This approach aimed to address the challenge posed by the large number of classes in the ImageNet dataset, specifically the potential bias towards matching the long tails of the soft target distributions. To evaluate this hypothesis, we compared the KL divergence between full soft targets and subsets of soft targets. By selecting the top-10 and top-100 most probable classes based on the teacher's predictions, we observed that some teacher-student pairs exhibited higher divergence over all classes compared to the selected subsets. This indicated the influence of classes with low prediction probabilities on the KL divergence.

Motivated by these findings, we further examined the impact of considering only the top-10 or top-100 classes on the transfer performance. Across six teacher-student pairs, using the top-10 divergence resulted in an average increase of 0.20 percentage points in transfer delta. Moreover, we observed that the magnitude of improvements aligned with the differences between the top-10 and total KL divergence. Our findings suggest that limiting the divergence to selected classes can be advantageous when dealing with a large number of classes, although the magnitude of improvements remains limited.

**Contrastive distillation for knowledge transfer between arbitrary models** To understand how well contrastive distillation techniques are suited for knowledge transfer between arbitrary

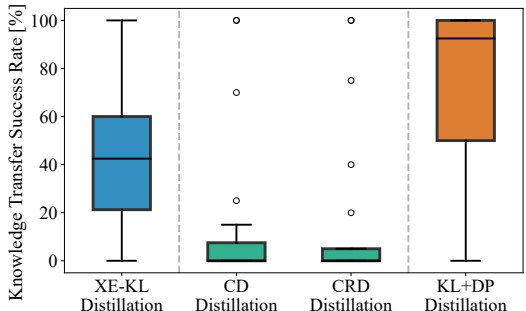

Figure 6: Share of teacher increasing student performance (success rate) for contrastive distillation (green) vs classification-guided distillation (blue) and continual learning based KL+DP (orange).

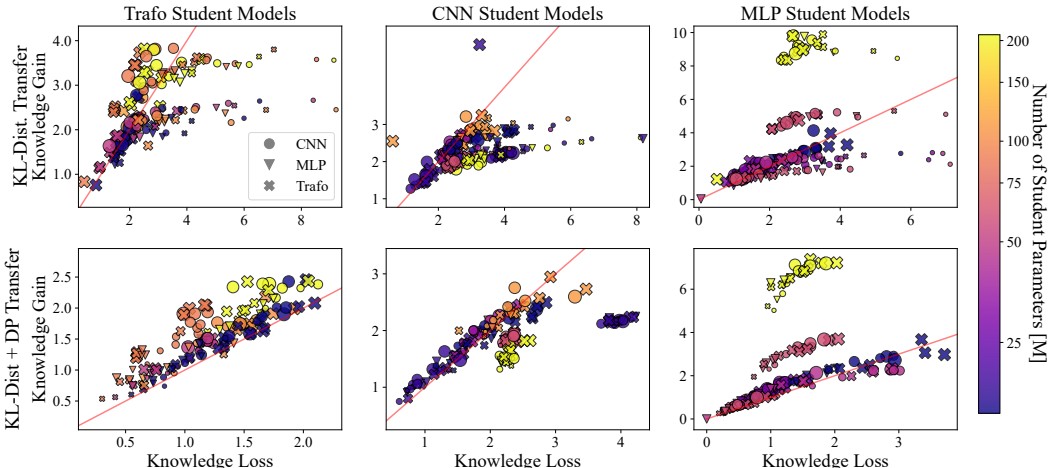

Figure 7: Share of transferred knowledge (knowledge gain) visualized against the share of knowledge lost for vanilla KL distillation and our proposed KL+DP distillation approach. Student models are grouped by their respective architecture type. Each marker represents one teacher-student pair. The color of the markers represents the size of the student, while marker shapes determine the teacher architecture. The marker size visualizes the teacher's performance. Results showcase a clear benefit of KL+DP, moving most points to areas of positive knowledge transfer (above red diagonal).

pretrained models, we measure the average transfer success rate for both CD and CRD distillation transfer (§A.2), with results shown in Fig. 6. We leverage the same experimental setup on 10% ImageNet as for the other transfer approaches (see §5.1). The experimental results clearly show the contrastive distillation approaches to be unable to improve the student model for most teacher models. On closer examination of the results we can see that the contrastive distillation approaches result in similar levels of knowledge transfer from the teacher to the student, but appear to also incur much stronger overall overwriting, causing the student to lose large portions of its previous knowledge. While very suitable for distillation to untrained students, this behaviour is unfortunately not well suited for knowledge transfer between already trained expert models.

## B.2 Extended Results on Knowledge Transfer Between Pretrained Models

For our knowledge transfer success rate experiments conducted in Section 5.1, we provide an extended and more detailed version for Figure 4a in Figure 7. Using a scatterplot, we relate the share of knowledge transferred to the student model (knowledge gain) versus the share of the students pretrained knowledge that is overwritten during the transfer process (knowledge loss). Each student model is denoted by a respective color choice associated with its parameter count. Symbol sizes and colors denote both family and performance of the respective teacher models. The red line denotes an equal trade-off between knowledge gain and loss, with upper-diagonal entries indicating a positive knowledge transfer. Comparing the results of vanilla *KL-Dist. transfer* and the continual learning based *Kl+DP transfer*, we see that a vast majority of points are pushed up the diagonal, allowing for transfer even from weaker models (small symbols, heavily scattered towards the lower diagonal area in the normal knowledge distillation approach). This behaviour also highlights that normal knowledge distillation approaches generally overwrite knowledge instead of augmenting, and is reflected in our correlation studies in Figure 4a.

Overall, these results simply extend the insights provided in the main part of this work from a more detailed point of view, highlighting that a continual learning treatment of the knowledge transfer problem can significantly raise the transfer success rate. However, we note that this more finegrained perspective *provides better support on the detrimental aspect of stronger visual inductive biases for general knowledge transfer*, as we found CNN students to generally perform worst, even when leveraging *KL+DP transfer*.

Table 7: Knowledge Transfer results on full ImageNet, from four teacher to eight selected student models. The tables include the individual transfer deltas of all teacher-student pairs.

| | $\Delta_{transf.}$ KL-Dist. | | | | $\Delta_{transf.}$ KL-Dist.+DP | | | | $\Delta_{transf.}$ KL-Dist.+DP (unsup.) | | | |
|---|---|---|---|---|---|---|---|---|---|---|---|---|
| Teachers → 
 ↓ Students | SWSL- 
 ResNext101 | Volo-D2 | ResMLP36 | CoaT- 
 lite-mini | SWSL- 
 ResNext101 | Volo-D2 | ResMLP36 | CoaT- 
 lite-mini | SWSL- 
 ResNext101 | Volo-D2 | ResMLP36 | CoaT- 
 lite-mini |
| XCiT-P16 | 0.95 | 1.40 | -0.55 | -0.88 | 0.93 | 0.90 | 0.36 | 0.42 | 1.29 | 1.45 | 0.57 | 0.52 |
| PiT-B | 1.16 | 1.42 | -0.24 | -0.74 | 0.74 | 0.86 | 0.31 | 0.31 | 1.35 | 1.59 | 0.43 | 0.45 |
| PiT-XS | 0.54 | 0.43 | 0.14 | -0.71 | 0.55 | 0.44 | 0.37 | 0.23 | 0.51 | 0.53 | 0.29 | 0.08 |
| SeNet154 | 0.38 | -0.07 | -0.20 | -0.32 | 0.38 | 0.02 | 0.22 | 0.48 | 0.42 | 0.13 | 0.09 | 0.38 |
| ConvNext | 0.23 | 0.41 | -1.10 | -1.57 | 0.49 | 0.44 | 0.26 | 0.12 | 0.44 | 0.38 | 0.22 | 0.09 |
| ResNetV2 | 0.34 | 0.11 | -0.23 | -0.56 | 0.32 | 0.17 | 0.28 | 0.13 | 0.34 | 0.18 | 0.29 | 0.11 |
| Mixer-B16 | 0.32 | 0.22 | -0.64 | -1.07 | 0.31 | 0.22 | 0.11 | -0.05 | 0.35 | 0.26 | 0.07 | -0.02 |
| ResMLP-24 | 0.58 | 0.43 | -0.16 | -0.26 | 0.57 | 0.45 | 0.10 | 0.20 | 0.57 | 0.36 | 0.10 | 0.13 |

Table 8: The table below shows the results of knowledge transfer with our proposed KL-Dist. + DP transfer approach on the full ImageNet. It includes two metrics that describe the changes in the positive and negative prediction flips, and extends the information provided in Table 1. For each student, we report the mean and standard deviation over all teacher models, which can be found in Table 6.

| Students | Type | Acc. | # Param. | $\Delta_{transf.}$ | $\Delta_{\rho^{pos}}$ | $\Delta_{\rho^{neg}}$ |
|---|---|---|---|---|---|---|
| XCiT-P16 (El-Nouby et al., 2021) | Trafo | 82.89 | 189.10 | **0.65** ($\pm0.26$) | -0.74 ($\pm0.25$) | -0.08 ($\pm0.04$) |
| PiT-B (Heo et al., 2021) | Trafo | 82.44 | 73.76 | **0.55** ($\pm0.25$) | -0.64 ($\pm0.28$) | -0.08 ($\pm0.05$) |
| PiT-XS (Heo et al., 2021) | Trafo | 78.19 | 10.62 | **0.40** ($\pm0.12$) | -0.45 ($\pm0.13$) | -0.05 ($\pm0.04$) |
| SeNet154 (He et al., 2018) | CNN | 81.23 | 115.09 | **0.27** ($\pm0.17$) | -0.35 ($\pm0.14$) | -0.07 ($\pm0.04$) |
| ConvNext (Liu et al., 2022) | CNN | 84.57 | 50.22 | **0.33** ($\pm0.14$) | -0.37 ($\pm0.09$) | -0.04 ($\pm0.06$) |
| ResNetV2 (He et al., 2016b) | CNN | 82.80 | 25.55 | **0.23** ($\pm0.08$) | -0.35 ($\pm0.09$) | -0.13 ($\pm0.03$) |
| Mixer-B16 (Tolstikhin et al., 2021) | MLP | 82.30 | 59.88 | **0.15** ($\pm0.13$) | -0.16 ($\pm0.08$) | -0.01 ($\pm0.07$) |
| ResMLP-24 (Touvron et al., 2021) | MLP | 80.76 | 30.02 | **0.33** ($\pm0.19$) | -0.30 ($\pm0.16$) | +0.03 ($\pm0.04$) |

The following table shows the individual transfer deltas of the teacher-student pairs from Table 1 and Table 3.

To further support our analysis in Section 5.1, we have provide additional results regarding the change in the share of positive and negative prediction flips during knowledge transfer. Positive prediction flips $\rho^{pos}$ refer to cases where the teacher was correct, but the student was incorrect. In contrast, negative prediction flips $\rho^{neg}$ refer to cases where the teacher was incorrect, but the student was correct. To measure this change, we defined two new metrics, pos-flips delta $\Delta_{\rho^{pos}}$ and neg-flips delta $\Delta_{\rho^{neg}}$, similar to the transfer delta. We present the mean and standard deviation for both metrics for all student models using our *KL+DP transfer* approach in Table 8, extending the results from Table 1.

Our goal with knowledge transfer is to transfer complementary knowledge, i.e., the positive prediction flips. This means that the number of samples where the teacher is correct but the student is incorrect should decrease as much as possible. However, we must simultaneously preserve the student's previous knowledge. As a result, the number of samples where the student is correct and the teacher is incorrect (negative prediction flips) should not decrease.

The experimental results conclusively demonstrate the effectiveness of our approach in reducing the share of positive prediction flips for all student models. This underlines the capability of our approach to transfer complementary knowledge between models. Moreover, the minor changes in the negative prediction flips provide compelling evidence of the approach's ability to preserve the student's previous knowledge.

## B.3 EXTENDED RESULTS ON THE IMPACT OF DIFFERENT STUDENT MODEL PROPERTIES ON KNOWLEDGE TRANSFER

In this section, we provide a closer assessment of the impact of the student model properties on the knowledge transfer behaviour, as measured through the transfer delta. In particular, we look at performance, size (measured by the number of parameters) and model family. For this assessment, we selected for each model property pairs or triplets of students with similar values for two of the three properties to isolate each single variable as well as possible. While an exact intervention can not be made by simply leveraging pretrained models, this setup does provide more controlled insights, which we visualize in Figure 8 for experiments conducted on 10% ImageNet using the *KL+DP transfer* approach.

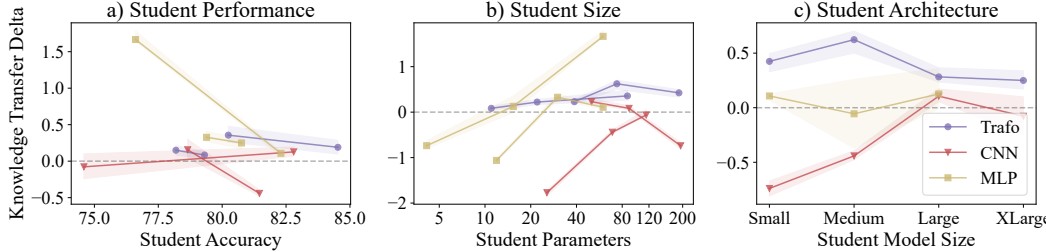

Figure 8: Evaluation of the impact of the student model properties a) performance, b) size (measured by the number of parameters) and c) architecture type on the knowledge transfer delta. Each marker represents a selected student model distilled with 20 different teacher models. We group students into pairs or triplets based on the remaining model properties by connecting the respective markers.

Note that each marker represents *one evaluated student model with all 20 teacher models*. We connect the pairs or triples of students that can be compared, with the color of the lines and markers representing the model family of the student.

Our results replicate insights noted in the main part of this work, particularly Figure 5 (right). We find that even when controlling for other factors such as initial accuracy, the overall student capacity appears strongly correlated with the ability to receive new knowledge without overwriting previous. This is a particularly pronounced behavior in models with strong visual inductive bias such as CNNs. The rightmost subfigure showcases that when looking at the average behavior of a model family (divided into different model sizes), that scale can offer emergent transfer capabilities in CNNs - while not available before - for any type of specific architecture - increased sizes can allow for notably improved transferability.

## B.4 Extended results on additional datasets

To substantiate our results on ImageNet we additionally conduct experiments on the CUB200 Wah et al. (2011), Caltech256 Griffin et al. (2007), and Stanford-Cars Krause et al. (2013) datasets.

For each datasets we combine the nine student and four teacher models as shown in Table 6 resulting in a total of 36 teacher-student combination. We fine-tune the classification layer of the student and teacher models using dataset-specific data before initiating knowledge transfer. We employ the dataset's training data as the transfer set.

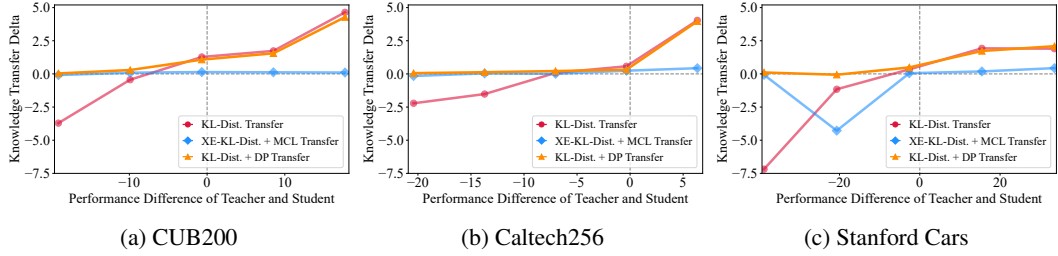

|  (a) CUB200 | (b) Caltech256 | (c) Stanford Cars |

Figure 9: Knowledge transfer delta based on teacher-student performance difference for three additional datasets: a) CUB200, b) Caltech256, and c) Stanford Cars. We compare simple *KL-Dist.* transfer with *XE-KL-Dist.+MCL* transfer and *KL-Dist.+DP* Transfer. The teacher-student pairs are categorized into bins determined by equipartitions of their respective performance differences. To mitigate the influence of outliers, we report the mean transfer delta of the top 25% within each bin and approach.

Across all datasets, we consistently observe the *KL-Dist.+DP transfer* approach to not only enable the transfer of knowledge from less proficient teachers without compromising student performance but to also demonstrate the capacity to transfer substantial knowledge portions in cases where the teacher outperforms the student significantly, aligning with the effectiveness of the straightforward *KL-Dist. transfer*. These results are in line with our observations on ImageNet (c.f. Figure 4b) and underline the strengths of *KL+DP transfer*.

## B.5 EXTENDED RESULTS ON KNOWLEDGE TRANSFER UNDER DOMAIN SHIFTS

We further explore knowledge transfer in the setting of a domain shift between the teacher and student model. For this purpose we fine tune the teacher model on the domainnet infograph dataset Peng et al. (2019) before conducting knowledge transfer. The transfer process is executed on the 10% subset of ImageNet. Our comprehensive assessment encompasses a cohort of 9 distinct student models and 4 teacher models (see Table 6).

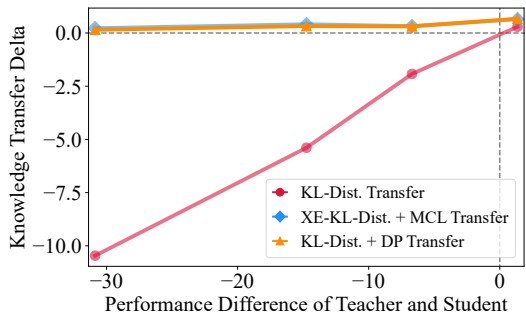

Figure 10: Inter-domain knowledge transfer delta analysis for *KL-Dist.* and *KL-Dist.+DP* transfer. We investigate the transfer delta resulting from knowledge transfer from a teacher model trained on DomainNet Infograph to an ImageNet-pretrained student model.

Notably, our findings underscore the efficacy of the *KL-Dist.+DP transfer* approach, which facilitates the transfer of knowledge from the Infograph-trained teacher to the student model on the ImageNet domain, thereby improving the student's performance. In stark contrast, the conventional *KL-Dist. transfer* demonstrates a substantial decrease in student accuracy, particularly when using a less proficient teacher.

## B.6 EXTENDED RESULTS ON THE TRANSFER FROM MULTIPLE TEACHERS

Finally, we present additional insights on the sequential knowledge transfer from multiple teacher models to a single pretrained student model. For all multi-teacher knowledge transfer experiments we select three student models (XCiT-P16, Twins, PiT-B) and three teacher models (SWSL-ResNext101, VOLO-D2, ResMLP-36) from Tab. 6. Appendix B.6 visualizes the knowledge transfer (knowledge gain), the share of the student's pretrain knowledge being lost (knowledge loss) and the overall transfer delta over the transfer epochs for the PiT-B Heo et al. (2021) student model presented in §5.2. As noted there, we distill the student with three different teacher models (see Table 6). For this particular visualization, we order teachers by ascending performance, but find positive continual transfer also to be achievable from

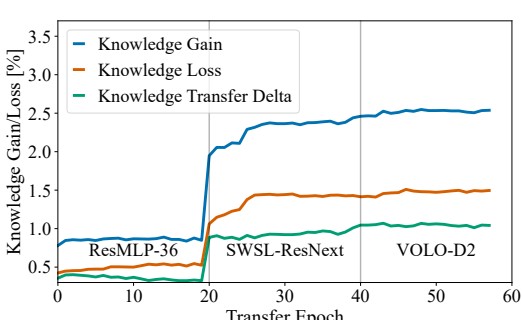

Figure 11: Knowledge transfer (knowledge gain) and loss of the student previous knowledge (knowledge loss) during the sequential training of PiT-B Heo et al. (2021) with three different teacher models sorted by ascending performance.

other sequences. For each teacher, we allocate a fixed transfer budget of 20 epochs. As noted already in Table 2, the figure visually highlights that positive transfer deltas can be gained going from one teacher to the subsequent one (stronger transfer delta compared to the strongest single student, $\Delta_{\text{dist}} = 1.04$), but with returns diminishing. We can attribute this to the increased rate of forgetting - while knowledge gain is steadily rising, continuously moving the student from its initial pretraining weights induces increasingly stronger knowledge loss, even when leveraging *Kl+DP transfer*.

For further insights, we compare the results of our multi-teacher experiments using *KL-Dist.+DP transfer* to vanilla *KL-Dist. transfer* (Tab. 9). The results clearly show that sequential *KL-Dist.*

Table 9: *Knowledge transfer from multiple teachers* into a pretrained student using sequential, and soup-based vanilla KL-Dist. transfer (c.f. §4.3). We compare with transfer deltas obtained from the single teacher knowledge transfer.

| Students | Type | Acc. | # Param. | $\Delta_{transf.}$ Single Teacher | | | $\Delta_{transf.}$ Multiple Teachers | |
|---|---|---|---|---|---|---|---|---|
| | | | | Mean | Min | Max | Sequ. | Soup |
| XCiT-P16 El-Nouby et al. (2021) | Transf. | 82.89 | 189.1 | 0.47 | -0.85 | **1.31** | 0.48 | 0.89 |
| Twins Chu et al. (2021) | Transf. | 83.68 | 99.27 | -0.04 | -1.04 | **0.63** | 0.01 | 0.43 |
| PiT-B Heo et al. (2021) | Transf. | 82.44 | 73.76 | 0.69 | -0.24 | **1.39** | 0.80 | 1.19 |

*transfer* cannot achieve larger gains as the best teacher alone but results in performance gains in the range of the average transfer delta across the three teachers. This again shows that rather than transferring only the complementary knowledge vanilla *KL-Dist. transfer* overwrites the student's previous knowledge with the knowledge of the teacher model. Thus when sequentially transferring knowledge from multiple teachers improvements from the previous transfer are lost during transfer from the subsequent teacher. Note that the vanilla *KL-Dist. transfer* approach cannot be directly applied to transfer knowledge from multiple teacher models in parallel, hence we omit this baseline.

