# OpenReview forum: "Fantastic Gains and Where to Find Them: On the Existence and Prospect of General Knowledge Transfer between Any Pretrained Model"
_ICLR.cc/2024/Conference — ICLR 2024 spotlight_

### Official Review · Reviewer_yxq3 · 2023-10-28

**Soundness:** 4 excellent
**Presentation:** 4 excellent
**Contribution:** 4 excellent
**Rating:** 8
**Confidence:** 3

**Summary:**

The authors provide an empirical study of the ability to transfer complementary knowledge between different pretrained models without performance degradation. This paper analyzes existing approaches in knowledge distillation and find it insufficient, especially in the case of distilling specific information from weaker teacher models. They go on to propose a data partitioning-based method (into regions of desired teacher behavior and desired student behavior retention) to achieve complementary knowledge transfer between the pretrained models considered in this paper.

**Strengths:**

1. The paper empirically shows that complementary knowledge exists between a large suite of models defined by different architectures and sizes (and even in weaker models that are less well performant than other stronger models). This complementary knowledge is localized to particular classes (what the authors deem as relative areas of expertise)

2. The authors propose a new data-partitioning approach to transfer complementary knowledge, where data is partitioned by which model has a higher probability of the ground truth class (or chosen simply by maximum probability in an unsupervised case).

3. Extensive experimental results that demonstrate that the proposed distillation approach transfers at a higher rate and transfers complementary knowledge from weaker teacher models.

4. The paper also studies different properties of student models that better allow for knowledge transfer.

**Weaknesses:**

Overall, I think the paper is quite comprehensive. A few points that may be lacking:

1. The results in studying properties of student models is a bit surprising to me. This isn’t a huge weakness, but more exploration of why CNN student models improve with scale and why transformer student models seem to worsen with would strengthen these results.

2. The data partitioning heuristic is reasonable, but some ablations on this approach would be more enlightening. Perhaps in some instances, the student model may be overconfident about particular data points (that either have incorrect labels or are inherently difficult examples to classify), and this data partitioning approach would maintain this overconfidence.

**Questions:**

1. Do you have any intuitions as to why student models that are CNNs exhibit better transfer at scale (while other architectures do not)? (Figure 8 in Supplement)

2. In Table 3, unsupervised DP outperforms supervised DP on several tasks. This seems a bit surprising; in the cases where these methods would be different, your DP approach would be distilling information from the teacher model on instances where the model is both quite confident and incorrect. Do you have an ideas about how this would be beneficial, and does this match your intuitions as to why this method works in general?

---

> ### Author Response · Authors · 2023-11-14
> **Response to Review**
>
> We thank the reviewer for their positive and useful review, and address each issue & question raised individually below.
>
> ---
>
> __Issue 1__
> > The data partitioning heuristic is reasonable, but some ablations on this approach would be more enlightening. Perhaps in some instances, the student model may be overconfident about particular data points (...), and this data partitioning approach would maintain this overconfidence.
>
> The KL-Dist.+DP transfer approach __does not introduce additional hyperparameters compared to vanilla KL-Distillation__, with robustness towards e.g., temperature choices (see Supp. A.3) - as such, we believe the most essential method ablations to be covered. In addition, we provide comparisons between supervised and unsupervised partitioning in Section 5.1 and Table 3, which showcase comparable performance but at the cost of increased transfer variance in the unsupervised case.
>
> As the reviewer correctly points out, the confidence-based partitioning heuristic is imperfect due to issues such as overconfidence, which likely plays into this issue - and while the final results showcase comparable performance, some additional insights could be beneficial.
>
> Consequently, to assess the partitioning of the input data, we calculate the number of positive and negative flip samples from the current training batch that are transferred to the teacher model. We use two metrics, namely “Pos. to Teacher” and “Neg. to Teacher”.
> Ideally, the positive flip samples, where the teacher model is correct while the student model is incorrect, should be assigned to the teacher model (i.e., Pos. to Teacher = 1). On the other hand, the negative flip samples, where the student model is correct and the teacher model is incorrect, should not be assigned to the teacher model (i.e., Neg. to Teacher = 0). We evaluated the partitioning heuristic for the teacher and student models presented in Table 6 using our KL-Dist. + DP Transfer approach.
>
> Our analysis shows that, on average, 72% of the positive flip samples and 9% of the negative flip samples were assigned to the teacher model.
> This means that while the simple confidence-based partitioning heuristic does not perfectly assign the training samples, it does align with the idea of partitioning the training data based on the teachers' and students' areas of expertise. Our findings are consistent with the transfer delta results, where we observed that the KL+DP approach successfully transferred knowledge from the teacher while maintaining the students' pretrained knowledge. Nevertheless, we think the proposed approach would benefit from a more precise partitioning, and we hope to explore this further in future work. We will incorporate the results above in the final version of this paper.
>
> ---
>
> __Question 1__
> > Do you have any intuitions as to why student models that are CNNs exhibit better transfer at scale (while other architectures do not)? (Figure 8 in Supplement)
>
> In general, we observe that student models from all architecture families have a strong tendency to improve with an increase in size, as demonstrated in Figure 5b and Figure 8b. However, it is worth mentioning that it can be challenging to isolate the effect of the student model's size from the impact of other variables, such as the large number of potential model variations within a model family.
> Particularly for CNNs, we found that while transfer receptiveness increases with scale, these models are generally more prone to have their previous knowledge overwritten at reduced capacity levels (see Section 5.1, p.9 and supplementary).
> We currently attribute this to the stronger visual inductive biases inherently encoded into the model architecture, which limits the structure of knowledge that can be effectively incorporated. This hypothesis is reinforced by the behavior of the other model families. However, a more extensive study could provide further interesting insights, which we will leave to future research to investigate!
>
> ---
>
> __Question 2__
> > In Table 3, unsupervised DP outperforms supervised DP on several tasks. This seems a bit surprising (...). Do you have any ideas about how this would be beneficial, and does this match your intuitions as to why this method works in general?
>
> As noted by the reviewer, knowledge transfer with unsupervised KL+DP performs comparably and, in some cases, better than the supervised variant. This was more likely for stronger and larger teacher models (positive correlation, ~0.25, between teacher model accuracy and the binary event of the unsupervised variant outperforming the supervised one). We can attribute this to the stronger teacher more consistently producing high confidence scores compared to the weaker student, leading to a favorable, closer approximation of the KL distillation transfer performance for high teacher-student performance differences (c.f. Fig. 4b). We will incorporate this information into the final version of this paper.

---

> > ### Comment · Reviewer_yxq3 · 2023-11-19
> > **Reviewer response**
> >
> > Thanks for the thorough responses! I appreciate the additional results that demonstrate the confidence-based heuristic works quite well (only 9% incorrect assignments). Overall, my questions/concerns are all addressed.

---

### Official Review · Reviewer_QCZm · 2023-11-01

**Soundness:** 4 excellent
**Presentation:** 4 excellent
**Contribution:** 4 excellent
**Rating:** 8
**Confidence:** 4

**Summary:**

This paper studies general knowledge distillation (KD), where, given any two models, the student can infer missing information from the teacher model and potentially improve the performance. The authors begin with a comprehensive analysis showing that such complementary knowledge generally exists in any paired models regardless of model capacity or architecture, and existing KD methods cannot leverage the information that student models already carry, i.e., trained students. To this end, the authors propose a continual learning-based extension to the existing KD methods and a data partitioning scheme that, according to the highest prediction probability., simultaneously maintains the useful knowledge of students while learning from the teacher. The extensive experiments conducted on more than 400 models sufficiently verify the effectiveness and provide many insightful analyses.

**Strengths:**

1. The problem studied in this paper, i.e., general knowledge distillation, is interesting and practical. It bridges the gap in the existing literature that a trained student might degrade during the knowledge distillation process.

2. The additional analysis also indicates almost every model could benefit from the other teacher models, even if the teachers are weaker than the students. The result and the evaluation methodology may motivate the community for further research.

3. The proposed method is sound yet easy to implement in practice. The authors also consider different scenarios of distillation, including a single teacher and multiple teachers distilled in different ways/orders.

4. The large-scale analysis (over 400 models) validates the claim and provides various insights, such as the properties of student models.

**Weaknesses:**

1. The student models considered in this paper are relatively strong. As the authors claim **general** knowledge distillation, the readers will also be interested in the weaker models or even from scratch. However, the authors only consider powerful architectures, such as transformers or ResNet, in the paper.

2. The proposed method is slightly confusing to me. My understanding is that the proposed data partition is built upon the continual learning method since Figure 4(a) clusters KL-Dist + DP Transfer into continual learning. If so, the contribution of each component is not clear enough to me. Though some experiments, e.g., Figure 4(b), present the partition improves the MCL transfer, the contribution of the partition itself is not investigated in the experiments.

3 (Minor) Consistency of italic type. Some "KL+DP" are italics in the text, while some are not.

**Questions:**

1. What would happen if one applies the proposed method to weaker models, e.g., randomly initialized? I guess it will be degenerated to conventional KD methods. A specific section for weaker/smaller models would be interesting to the community of edge device users/researchers.

---

> ### Author Response · Authors · 2023-11-14
> **Response to Review**
>
> We appreciate the useful and positive feedback and have addressed each question raised individually.
>
> ---
>
> __Issue 1__
> > The student models considered in this paper are relatively strong. As the authors claim general knowledge distillation, the readers will also be interested in the weaker models or even from scratch. However, the authors only consider powerful architectures, such as transformers or ResNet, in the paper.
>
> While the reviewer is correct in that a larger part of the models available within the timm model library may be considered on the stronger end, we want to highlight that timm generally contains all sorts of models of practical interest. This includes smaller and “weaker” ones as well. We deliberately also included such weaker models in our study, as seen in Supp. Table 6 which contains models varying in both model sizes and performances, with performance gaps of up to 15% and tiny transformer models such as PiT-XS with only 11 Million parameters. __For all these models, we find complementary knowledge to clearly exist (see Fig. 1) and complementary knowledge transfer to still be possible__.
>
> For very weak or even randomly initialized student models, the setting would transform into a regular knowledge distillation problem. As in this scenario, there would be no benefit in retaining the student’s initial performance, our KL+DP approach would not be feasible.
> For randomly initialized teachers, we show in Section 3, p.4 that no complementary knowledge exists.
>
> However, we do agree that an investigation into well-to-moderately performant models of even smaller size or particular edge-device models could be of high relevance.
> As we find the model capacity to be a driving factor in the amount of complementary knowledge that can be received (see our small model size scaling laws study in Fig. 5b), finding ways to break this trend for edge-device models would make _a particularly interesting direction for future research to build on_!
>
> ---
>
> __Issue 2__
> > The proposed method is slightly confusing to me. My understanding is that the proposed data partition is built upon the continual learning method since Figure 4(a) clusters KL-Dist + DP Transfer into continual learning. If so, the contribution of each component is not clear enough to me. Though some experiments, e.g., Figure 4(b), present the partition improves the MCL transfer, the contribution of the partition itself is not investigated in the experiments.
>
> We thank the reviewer for the question and will first describe our proposed approach in more detail. In particular, for general knowledge transfer, KL-Dist. + DP Transfer effectively extends the standard vanilla KL-Dist. transfer setting. To protect the students' pretrained knowledge during the transfer, we introduce a “student-teacher” model (essentially the unaltered, initial pretrained student, see also Fig. 3).
> This borrows from replay regularization used in continual learning, where we utilize the student-teacher model to replay the students' previous knowledge.
>
> Our data partitioning scheme comes into play for selecting which samples we use for replay from the student's previous knowledge and for determining where to adapt to the teacher context - either leveraging supervision or utilizing confidence estimates for fully unsupervised transfer. Consequently, we consider the KL+DP approach inspired from the perspective of continual learning.
>
> Furthermore, we provide analyses into the partitioning behavior in Section 5.1, p.8, through our comparison between supervised and unsupervised partitioning in Table 3, where we discover comparable general transfer effectiveness, but at the cost of higher transfer variance (see first paragraph p.8).
>
> Finally, KL+DP does not introduce additional hyperparameters compared to vanilla KL-Distillation, with robustness towards e.g., temperature choices (see Supp. A.3) - as such, we believe essential method ablations that allow us to better understand our proposed method to be covered. We do include an additional experiment studying the sample assignments in unsupervised partitioning in our reply to Reviewer yxq3, where we find unsupervised partitioning to assign samples for the majority of cases correctly (Reply 1).
>
> Overall, we will make sure to better highlight the detailed method aspects as described above in the final version of the paper.
>
> ---
>
> __Issue 3__
> > Consistency of italic type. Some "KL+DP" are italics in the text, while some are not.
>
> We thank the reviewer for spotting this! We will ensure our notation typesetting to be fully consistent in the final version of the paper.

---

> ### Comment · Reviewer_QCZm · 2023-11-21
> **Response to the authors**
>
> I would like thank the authors for their good work and the response, which has addressed my concerns. I will keep my score and recommend acceptance.

---

### Official Review · Reviewer_epvt · 2023-11-06

**Soundness:** 3 good
**Presentation:** 3 good
**Contribution:** 3 good
**Rating:** 8
**Confidence:** 3

**Summary:**

This paper investigates the phenomenon that different models have complementary information, reflected in their different predictions on each sample. Such complementary information could be due to model architectures, training settings, etc. Then the authors study how to transfer the complementary knowledge from a teacher model to a student model. The authors formulate this as a continual learning problem, which effectively improves upon the knowledge distillation baseline  and achieves better knowledge distillation on diverse models and ImageNet.

**Strengths:**

1. This paper identifies the "complementary knowledge" in different neural networks with grounded evidence. I think the claims are reasonable and well-supported.

2. The authors proposed improvement to the general knowledge distillation approaches from the continual learning perspective, including constraining the weight updates and transfer data. Both of the approaches are reasonable and straightforward to apply.

3. The authors conduct experiments on a wide range of models on ImageNet and show improvement with their improved continual knowledge distillation approach.

**Weaknesses:**

1. Present the key terms more clearly. For example, I haven't found the definition of transfer delta $\Delta_{transf}$, which is an important evaluation metric.

2. I think the proposed regularization, including the constraining of weight updates and transfer data, are some tricks for knowledge distillation. Although I don't work directly in knowledge distillation and I will wait for other expert reviewers to justify the novelty, I think the authors need to clarify more about their motivation or form a more explicit connection with continual learning. I also raised a question in the next section, elaborating my concerns.

3. I suggest the authors add some more experiments to make the arguments more thorough. Specifically, a more detailed breakdown of the accuracy numbers would help. For instance, (a) the percentage of [teacher correct, student wrong] samples are changed to correct answers after the distillation, (b) the percentage of [teacher incorrect, student correct] samples are changed to incorrect labels, to understand if the transfer is indeed complementary.

**Questions:**

1. Explain more about the terms. The readers are likely to have an ambiguous understanding of them, including: transfer delta $\Delta_{transf}$, "available complementary knowledge per class," and transfer rate.

2. I am wondering if the methods have to reason from the "continual learning" perspective. In my opinion, the regularization techniques proposed by the authors seem like generally applicable tricks for knowledge distillation. If any procedure involving multiple steps has to be treated as continual learning, maybe training a neural network with SGD is also a continual learning process? I hope the authors can clarify this motivation better in the paper.

3. See the third weakness above.

4. I suggest the author add an oracle study to strengthen the argument. In the examples (e.g. Table 1), the final improvement seems small in scale. To argue that this is actually challenging, the authors can run several ensembles of the teacher-student model and compare it to the improvement from knowledge transfer. Of course, I welcome other variants of similar analytical studies from the authors.

---

> ### Author Response · Authors · 2023-11-14
> **Response to Review (1/3)**
>
> We thank the reviewer for their valuable and positive feedback and address each question raised individually.
>
> ---
>
> __Question 1__
> > Explain more about the terms. The readers are likely to have an ambiguous understanding of them, including: transfer delta, "available complementary knowledge per class," and transfer rate.
>
> We apologize for any confusion regarding the key terms mentioned in our paper. To clarify, the transfer delta, as defined in Section 4.1 on page 5, refers to the difference between the student model's accuracy before and after the knowledge transfer. This metric measures the improvement in top-1 accuracy achieved by transferring knowledge from the teacher to the student.
> Additionally, the term available complementary knowledge per class refers to the number of positive prediction flips within each class, as introduced in Section 3 on Page 4.
>
> Furthermore, in Section 5.1 on Page 8, we define the transfer rate as the student model's improvement in top-1 accuracy relative to the available complementary knowledge. In other words, the transfer rate indicates how much of the complementary knowledge is actually transferred to the student model.
>
> We understand that these definitions may have been insufficiently highlighted. We will improve and emphasize these for the final version through better visual separation and the above-provided additional explanations.
>
> ---
>
> __Question 2__
> > I am wondering if the methods have to reason from the "continual learning" perspective. In my opinion, the regularization techniques proposed by the authors seem like generally applicable tricks for knowledge distillation. If any procedure involving multiple steps has to be treated as continual learning, maybe training a neural network with SGD is also a continual learning process? I hope the authors can clarify this motivation better in the paper.
>
> Continual Learning generally comprises scenarios in which a model is exposed to context in a sequential, continual manner (e.g., images and associated class labels presented through a datastream). This means that a model will no longer have access to (and be able to train on) parts of this context after certain time- or training steps. As it continuously trains and learns new things, catastrophic forgetting of previous context knowledge occurs (c.f., e.g. [1,2,3]). This differs from regular SGD-style network training, where the same data can be accessed throughout the training process.
>
> In our particular case, we tackle the general knowledge transfer problem from the perspective of continual learning because crucial analogies can be drawn between both domains (see Section 4.2, p. 5-6): While transferring knowledge between trained models, we need to avoid catastrophic forgetting of previously learned feature context, while being able to incorporate complementary knowledge.
> This constraint is also significantly different from standard knowledge distillation, in which the idea and aspects of knowledge retention are not considered (see Section 4.1, p. 5). Consequently, standard knowledge distillation tools are unsuitable for general knowledge transfer between pretrained models, as can be seen experimentally, e.g., in Fig. 4a or 4b.
>
> However, we agree that this connection can be carved out more clearly, and we will incorporate the explanation above into the introductory text in Section 4.2.
>
> [1] Kirkpatrick et al. 2017, “Overcoming catastrophic forgetting in neural networks”
> [2] Zenke et al. 2017, “Continual Learning through Synaptic Intelligence”
> [3] Buzzega et al. 2020, “Dark Experience for Continual Learning: a Strong, Simple Baseline”

---

> > ### Author Response · Authors · 2023-11-14
> > **Response to Review (2/3)**
> >
> > __Question 3__
> > > I suggest the authors add some more experiments to make the arguments more thorough. Specifically, a more detailed breakdown of the accuracy numbers would help. For instance, (a) the percentage of [teacher correct, student wrong] samples are changed to correct answers after the distillation, (b) the percentage of [teacher incorrect, student correct] samples are changed to incorrect labels, to understand if the transfer is indeed complementary.
> >
> > We acknowledge the reviewer's suggestion of delving further into the performance gains achieved in our paper. To begin, we would like to first refer the reviewer to Figure 5a, page 8. Here, we have __analyzed the transfer rate in relation to the complementary knowledge available__. More specifically, we have selected subsets of classes accounting for the top-x% of the complementary knowledge (i.e., the classes with the highest share of positive prediction flips per class). For each of these subsets, we have evaluated the average transfer rate and have observed that the classes with the highest complementary knowledge have achieved the highest transfer rates. This observation __confirms that we transfer knowledge that we previously identified as complementary__.
> >
> > To further support our analysis, we have conducted __additional experiments to investigate the change in the share of positive and negative prediction flips during knowledge transfer__. Positive prediction flips refer to cases where the teacher was correct, but the student was incorrect. In contrast, negative prediction flips refer to cases where the teacher was incorrect, but the student was correct. To measure this change, we defined two new metrics, pos-flips delta and neg-flips delta, similar to the transfer delta. We present the mean and standard deviation for both metrics for all student models using our KL-Dist. + DP Transfer approach, extending the results from Table 1.
> >
> > __Table 1__
> >
> > | Students | transfer delta | pos-flips delta (lower better) | neg-flips delta (closer to zero better) |
> > |-------------|-------------|-------------|-------------|
> > | XCiT-P16 | 0.65 (+/-0.26) | -0.74 (+/- 0.25) | -0.08 (+/-0.04) |
> > | PiT-B | 0.55 (+/-0.25) | -0.64 (+/-0.28) | -0.08 (+/-0.05) |
> > | PiT-XS | 0.40 (+/-0.12) | -0.45 (+/-0.13) | -0.05 (+/-0.04) |
> > | SeNet154 | 0.27 (+/-0.17) | -0.35 (+/-0.14) | -0.07 (+/-0.04) |
> > | ConvNext | 0.33 (+/-0.14) | -0.37 (+/-0.09) | -0.04 (+/-0.06) |
> > | ResNetV2 | 0.23 (+/-0.08) | -0.35 (+/-0-09) | -0.13 (+/-0.03) |
> > | Mixer-B16 | 0.15 (+/-0.13) | -0.16 (+/-0.08) | -0.01 (+/-0.07) |
> > | ResMLP-24 | 0.33 (+/-0.19) | -0.30 (+/-0.16) | +0.03 (+/-0.04) |
> >
> > Our goal with knowledge transfer is to transfer complementary knowledge, i.e., the positive prediction flips. This means that the number of samples where the teacher is correct but the student is incorrect should decrease as much as possible. However, we must simultaneously preserve the student's previous knowledge. As a result, the number of samples where the student is correct and the teacher is incorrect (negative prediction flips) should not decrease.
> >
> > The experimental results __conclusively demonstrate the effectiveness of our approach in reducing the share of positive prediction flips for all student models__. This underlines the capability of our approach to transfer complementary knowledge between models. Moreover, the minor changes in the negative prediction flips provide compelling evidence of the approach's ability to preserve the student's previous knowledge.
> >
> > We will incorporate these additional results in the final version of this work.

---

> > > ### Author Response · Authors · 2023-11-14
> > > **Response to Review (3/3)**
> > >
> > > ---
> > >
> > > __Question 4__
> > > > I suggest the author add an oracle study to strengthen the argument. In the examples (e.g. Table 1), the final improvement seems small in scale. To argue that this is actually challenging, the authors can run several ensembles of the teacher-student model and compare it to the improvement from knowledge transfer. Of course, I welcome other variants of similar analytical studies from the authors.
> > >
> > > Our main objective in this study is to investigate the possibility of knowledge transfer __between__ any pair of pretrained models - as this allows the retention of a particular student model of choice without sacrifices or changes in speed, fairness, interpretability, or ease-of-use. __All of these properties are not retained in ensemble-based approaches (c.f. e.g., Section 1, p.2)__.
> > >
> > > Even more, we show that a general knowledge transfer mechanism allows for model gains from much weaker, lower-resource models as well, without the need for any external validation measures. As such (see also Section 1), insights from ensembling are __orthogonal__ to this work.
> > >
> > > However, to further highlight the last point, we also provide a simple experimental example in which we compare our KL-Dist. + DP transfer mechanism to ensembling in scenarios with architectural and performance differences between a student and a teacher.
> > > To ensure a more favorable setting for the ensembling baseline, we select two reasonably strong models with not too high of a performance difference: A ConvNeXt student at 84.57% ImageNet Top-1 accuracy and a ResMLP teacher at 80.75%. However, even here, our transfer can provide a positive transfer delta of +0.3%, while ensembling actually already drops performance by -0.2%. These differences are only exacerbated by taking a more flexible, higher-capacity student (as per our study in 5.1, p.9) and weaker teacher models.

---

> ### Author Response · Authors · 2023-11-22
>
> As the discussion period is closing soon, we would like to thank the reviewer again for their helpful feedback. We hope that our replies, additional results and our updated draft addressed all concerns, and that we better highlighted all the contributions included in this work. Of course, we are happy to continue the discussion in case there are any further questions.

---

### Official Review · Reviewer_LBqG · 2023-11-08

**Soundness:** 2 fair
**Presentation:** 3 good
**Contribution:** 2 fair
**Rating:** 5
**Confidence:** 4

**Summary:**

This paper investigates if it is possible to transfer complementary knowledge from one model to another without performance degradation. To this end, authors propose different heuristics to design how to switch the knowledge transfer between models. Experiments with various pairs of models demonstrate the effectiveness of the model-agnostic knowledge transfer.

**Strengths:**

1. The writing is clear and easy to follow.

2. There are consistent performance improvements compared to different types of baselines.

**Weaknesses:**

1. As addressed by the authors, different models are trained with different data augmentations, architectures and optimization techniques. The performance improvements are relatively marginal (e.g., Table 1), especially considering some models are not fully trained.

2. In Table 4, do the authors train all the variations with the same number of training steps? The sequential knowledge transfer may benefit from more training steps.

**Questions:**

See the weakness.

---

> ### Author Response · Authors · 2023-11-14
> **Response to Review**
>
> We thank the reviewer for their valuable feedback and address each question raised individually.
>
> Before doing that, we would just like to highlight that our paper is not just a simple benchmark comparison but rather a fundamental, proof-of-concept study on the prospects of general knowledge transfer (as also highlighted by the other reviewers), for which we not only propose an easy-to-utilize approach but provide the inherent initial motivation and support, alongside analyses on challenges and shortcomings.
>
> ---
>
> __Question 1__
> > As addressed by the authors, different models are trained with different data augmentations, architectures and optimization techniques. The performance improvements are relatively marginal (e.g., Table 1), especially considering some models are not fully trained.
>
> Our experiments do involve the use of models __fully pre-trained__ on ImageNet from the open-source model repository timm - these are extensively trained with hyperparameter tuning to optimize performance.
> At the same time, a gain of even a full percentage point on ImageNet-1k is generally considered significant - which we obtained _without any additional training data or optimization protocol changes and which holds even for already strong pretrained student models_.
>
> This is particularly relevant as we can achieve positive gains for __nearly all arbitrary model pairings__. In addition, the visualized model pairings were chosen by virtue of coverage of different model type pairings, as opposed to peak transfer performance, in order to show the generality of our proposed general knowledge transfer mechanism.
> However, as this work primarily serves as a proof-of-concept of both the existence and possibility of general knowledge transfer, we do agree that certainly further gains can be achieved, which we leave to future research to tackle.
>
> ---
>
> __Question 2__
> > In Table 4, do the authors train all the variations with the same number of training steps? The sequential knowledge transfer may benefit from more training steps.
>
> All experiments, including those in Table 4, use a fixed budget of 20 epochs to transfer teacher knowledge to the student. We have chosen this fixed budget as a practical estimate over which transfer generally converges. However, we have found that this is not the primary factor driving performance gains in the multi-teacher setting. As shown in Fig. 11, the student's performance improves within the first few epochs of transfer with each teacher model and quickly converges after that.
>
> In the sequential multi-teacher setting, this fixed transfer budget applies to each transfer step, which increases the overall number of transfer steps.
> In this study, we also tested fixed global numbers of transfer steps equal to the budget in the single-teacher scenario and experimented with increasing the number of transfer steps in the single-teacher setting. However, we found that these changes did not affect the observations reported in our paper.
>
> We will highlight these additional details, as described above, in the final version of this paper.

---

> ### Author Response · Authors · 2023-11-22
>
> As the discussion period is closing soon, we would like to thank the reviewer again for their helpful feedback. We hope that our reply and our updated draft addressed all concerns, and that we better highlighted all the contributions included in this work. Of course, we are happy to continue the discussion in case there are any further questions.

---

### Author Response · Authors · 2023-11-14
**General Comment**

We would like to thank all reviewers for their time and efforts to provide detailed and informative feedback, and their appreciation of the writing (_LBqG_), and overall excellent soundness, presentation, and contribution (_QCZm,yxq3_).

We particularly thank the reviewers for acknowledging our interesting and practical discovery of the general knowledge transfer problem bridging the gap in existing literature (_QCZm_), our novel, sound, and easy-to-implement approach to tackle this problem (_epvt,QCZm,yxq3_) which achieves consistent performance improvements for almost any model (_LBqG,QCZm_), and the overall significant and extensive experimental support (_epvt,QCZm,yxq3_).

We address each raised issue and question with individual replies for each reviewer.

---

A revision has also been uploaded to incorporate all mentioned changes, in particular
* Information about transfer budgets and better highlighting of the full approach (see also responses to _LBqG_, _QCZm_).
* Better highlighting the connection to Continual Learning (see _epvt_)
* Better highlight and describe definitions (as also explained in response to _epvt_).
* Additional results to investigate the change in the share of positive and negative prediction flips during knowledge transfer, and conclusively demonstrate the effectiveness of our proposed KL+DP approach in reducing the share of positive prediction flips for all student models (_epvt_, _yxq3_).
* Ensured consistent notation typesetting. (_QCZm_)
* Highlight that we do not need additional hyperparameters, and are robust to particular changes (_yxq3_).
* Provide additional context as to why unsupervised transfer works well, with additional study on the behaviour differences between the supervised and unsupervised variant (_yxq3_).

---

### Meta-Review · Area_Chair_vHx3 · 2023-12-12

**Metareview:**

This paper demonstrates the existence of complementary knowledge between different pretrained models, as evidenced by their distinct predictions on each sample, and further investigates the potential for transferring this knowledge without performance degradation. Given the limitations of existing knowledge distillation approaches, the authors frame the problem as a continual learning task and introduce a data partitioning scheme to maintain the useful knowledge of the student while learning from a teacher model. Empirical results show the effectiveness of the proposed approach across a variety of pretrained model pairs, including both strong and weak models. While primarily serving as a proof-of-concept, the study suggests the potential for general knowledge transfer, providing valuable insights for future research.

The reviewers generally acknowledge the paper's contribution in identifying and further transferring complementary knowledge between pretrained models, considering it both novel and solid. The proposed approach, viewed from a continual learning perspective, appears to be general, demonstrating effectiveness across a wide range of models on ImageNet. Notable and consistent performance improvements are observed. In the original manuscript, some terminology definition, implementation and experimental details, design rationale, and useful analyses (e.g., performance breakdown, difference between supervised and unsupervised partitioning) were missing or unclear. The authors’ responses, along with additional experimental results, have successfully addressed these concerns raised by the reviewers.

Three out of four reviewers recommend acceptance. Although the remaining reviewer did not provide follow-up feedback on the authors' response, it appears that the concerns raised have been addressed. I support the reviewers' recommendation. I suggest that the authors carefully consider the main points raised in the reviews, paying particular attention to additional analyses and experiments, when preparing the final version.

**Justification For Why Not Higher Score:**

The current recommendation is based on the reviewers’ ratings. A higher score seems to be not well justified.

**Justification For Why Not Lower Score:**

Three out of four reviewers recommend acceptance. Although the remaining reviewer did not provide follow-up feedback on the authors' response, it appears that the concerns raised have been addressed. I support the reviewers' recommendation. I suggest that the authors carefully consider the main points raised in the reviews, paying particular attention to additional analyses and experiments, when preparing the final version.

---

### Decision · Program_Chairs · 2024-01-16

Accept (spotlight)